# Interpretable Prototype-based Graph Information Bottleneck

**Sangwoo Seo**[1]  **Sungwon Kim**[1]  **Chanyoung Park**[1]*
[1]KAIST
{sangwooseo@kaist.ac.kr, swkim@kaist.ac.kr, cy.park@kaist.ac.kr}

## Abstract

The success of Graph Neural Networks (GNNs) has led to a need for understanding their decision-making process and providing explanations for their predictions, which has given rise to explainable AI (XAI) that offers transparent explanations for black-box models. Recently, the use of prototypes has successfully improved the explainability of models by learning prototypes to imply training graphs that affect the prediction. However, these approaches tend to provide prototypes with excessive information from the entire graph, leading to the exclusion of key substructures or the inclusion of irrelevant substructures, which can limit both the interpretability and the performance of the model in downstream tasks. In this work, we propose a novel framework of explainable GNNs, called interpretable **P**rototype-based **G**raph **I**nformation **B**ottleneck (PGIB), that incorporates prototype learning within the information bottleneck framework to provide prototypes with the key subgraph from the input graph that is important for the model prediction. This is the first work that incorporates prototype learning into the process of identifying the key subgraphs that have a critical impact on the prediction performance. Extensive experiments, including qualitative analysis, demonstrate that PGIB outperforms state-of-the-art methods in terms of both prediction performance and explainability. The source code of PGIB is available at https://github.com/sang-woo-seo/PGIB.

## 1  Introduction

With the success of Graph Neural Networks (GNNs) in a wide range of deep learning tasks, there has been an increasing demand for exploring the decision-making process of these models and providing explanations for their predictions. To address this demand, explainable AI (XAI) has emerged as a way to understand black-box models by providing transparent explanations for their predictions. This approach can improve the credibility of models and ensure transparency in their decision-making processes. As a result, XAI is actively being used in various applications, such as medical, finance, security, and chemistry [3, 26].

In general, explainability can be viewed from two perspectives: 1) ***improving the interpretability*** by providing explanations for the model's *predictions*, and 2) ***providing the reasoning process*** behind the model prediction by giving explanations for the model's *training process*. *Improving the interpretability* in GNNs involves detecting important substructures during the inference phase, which is useful for tasks such as identifying functional groups (i.e., important substructures) in molecular chemistry [29, 33, 23, 12]. On the other hand, it is also important to *provide the reasoning process* for why the model predicts in a certain way, which requires an in-depth analysis of the training phase, so as to understand the model in a more fundamental level. Through this reasoning process, we can visualize and analyze how the model makes correct or incorrect decisions, thus obtaining crucial information for improving its performance.

---

*Corresponding author.

37th Conference on Neural Information Processing Systems (NeurIPS 2023).

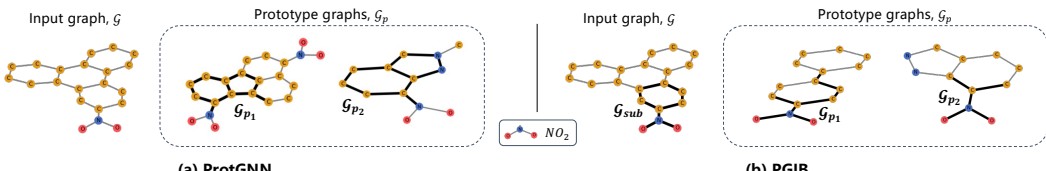

Figure 1: Comparison of the learned prototypes between ProtGNN and PGIB.

In recent years, there has been a growing interest in exploring the reasoning process to provide greater transparency and explainability in deep learning models. These approaches can be generally classified into two main categories: 1) ***post-hoc*** approaches, and 2) ***built-in*** approaches. *Post-hoc* approaches focus on exploring the model outputs by visualizing the degree of activation of neurons through measuring their contribution based on gradients of the model predictions. For instance, recent works [14, 9, 34] use techniques such as saliency maps and class activation maps to visualize the activated areas during the model's prediction process. However, these approaches require a separate explanatory model for each trained model, resulting in the need for a new explanatory model for additional training data and different models [2, 15]. In order to address the aforementioned challenges, *built-in* approaches aim to integrate the generation of explanations into the model training process. One such approach is prototype learning, which involves learning prototypes that represent each class of the input data, which are then compared with new instances to make predictions. ProtGNN [35], for instance, measures the similarity between the embedding of an input graph and each prototype, providing explanations through the similarity calculation and making predictions for the input graph based on its similarity with the learned prototypes. More precisely, ProtGNN projects each learned prototype onto the closest training graph, enabling it to provide explanations of primary structures for its prediction.

However, since ProtGNN compares the graph-level embedding of an input graph with the learned prototypes, the model overlooks the key substructures in the input graph while also potentially including uninformative substructures. This not only results in a degradation of the interpretability of the reasoning process, but also limits the performance on the downstream tasks. Figure 1(a) shows the prototype graphs in the training set (i.e., $\mathcal{G}_p$, denoted as bold edges) detected by ProtGNN for an input molecule (i.e., $\mathcal{G}$) that belongs to the "mutagenic" class. Despite the $NO_2$ structure being the key functional group for classifying a given molecule as "mutagenic," $\mathcal{G}_p$ detected by ProtGNN tends to include numerous ring structures (i.e., uninformative substructures) that are commonly found throughout the input graph, and exclude $NO_2$ structures (i.e., key substructures) in learned prototype graphs, which is mainly due to the fact that the input graph $\mathcal{G}$ is considered in the whole graph-level. As a result, it is crucial to identify a key subgraph within the input graph that holds essential information for the learning of prototypes, which in turn enhances both the explanation of the reasoning process and the performance on the downstream tasks. Among the various solutions for detecting important subgraphs, the Information Bottleneck (IB) has emerged as one of the most effective methods [27, 31], and it has been demonstrated that key subgraphs detected based on IB can contribute to performance improvement in various tasks such as relational learning [11] and structure learning [19]. We aim to approach the IB principle from the perspective of prototypes to convey important substructure information to the prototypes.

To this end, we propose a novel framework of explainable GNNs, called Interpretable **P**rototype-based **G**raph **I**nformation **B**ottleneck (PGIB). The main idea is to incorporate prototype learning within the Information Bottleneck (IB) framework, which enables the prototypes to capture the essential key subgraph of the input graph detected by the IB framework. Specifically, PGIB involves prototypes (i.e., $\mathcal{G}_p$) in a process that maximizes the mutual information between the learnable key subgraph (i.e., $\mathcal{G}_{sub}$) of the input graph (i.e., $\mathcal{G}$) and target information (i.e., $Y$), which allows the prototypes to interact with the subgraph. This enables the learning of prototypes $\mathcal{G}_p$ based on the key subgraph $\mathcal{G}_{sub}$ within the input graph $\mathcal{G}$, leading to a more precise explanation of the reasoning process and improvement in the performance on the downstream tasks. To the best of our knowledge, this is the first work that combines the process of optimizing the reasoning process and interpretability by identifying the key subgraphs that have a critical impact on the prediction performance. In Figure 1(b), PGIB is shown to successfully detect the key subgraph $\mathcal{G}_{sub}$ that includes $NO_2$ from the input graph $\mathcal{G}$, even when the ring structures are dominant in $\mathcal{G}$. It is important to note that PGIB is highly efficient in detecting $\mathcal{G}_{sub}$ from $\mathcal{G}$ since PGIB adopts a learnable masking technique, effectively resolving the time complexity issue. Last but not least, since the number of prototypes for each

class is determined before training, some of the learned prototypes may share similar semantics, which negatively affects the model interpretability for which the small size and low complexity are desirable [6, 25, 17]. Hence, we propose a method for effectively merging the prototypes, which in turn contributes to enhancing both the explanation of the reasoning process and the performance on the downstream tasks.

We conducted extensive experiments to evaluate the effectiveness and interpretability of the reasoning process of PGIB in graph classification tasks. Our results show that PGIB outperforms recent state-of-the-art methods, including existing prototype learning-based and IB-based methods. Moreover, we evaluated the ability of PGIB in capturing the label information by evaluating the classification performance using only the detected subgraph $\mathcal{G}_{sub}$. We also conducted a qualitative analysis that visualizes the subgraph $\mathcal{G}_{sub}$ and prototype graph $\mathcal{G}_p$, suggesting the ability of PGIB in detecting the key subgraph. Overall, our results show that PGIB significantly improves the interpretability of both $\mathcal{G}_{sub}$ and $\mathcal{G}_p$ in the reasoning process, while simultaneously improving the performance in downstream tasks.

In summary, our main contributions can be summarized as follows: **1)** We propose an effective approach, PGIB, that not only improves the interpretability of the reasoning process, but also the overall performance in downstream tasks by incorporating the prototype learning in a process of detecting key subgraphs based on the IB framework. **2)** We provide theoretical background with our method that utilizes interpretable prototypes in the process of optimizing $\mathcal{G}_{sub}$. **3)** Extensive experiments, including qualitative analysis, demonstrate that PGIB outperforms state-of-the-art methods in terms of both prediction performance and explainability.

## 2 Preliminaries

In this section, we introduce notations used throughout the paper followed by the definitions of IB and IB-Graph.

**Notations.** We use $\mathcal{G} = (\mathcal{V}, \mathcal{E}, \mathbf{A}, \mathbf{X})$ to denote a graph, where $\mathcal{V}$, $\mathcal{E}$, $\mathbf{A}$ and $\mathbf{X}$ denote the set of nodes and edges, the adjacency matrix and node features, respectively. We assume that each node $v_i \in \mathcal{V}$ is associated with a feature vector $\mathbf{x}_i$, which is the $i$-th row of $\mathbf{X}$. We use $\{(\mathcal{G}_1, y_1), (\mathcal{G}_2, y_2), \cdots, (\mathcal{G}_N, y_N)\}$ to denote the set of $N$ graphs with its corresponding labels. The graph labels are given by a set of $K$ classes $\mathcal{C} = \{1, 2, \ldots, K\}$, and the ground truth label of a graph $\mathcal{G}_i$ is denoted by $y_i \in \mathcal{C}$. We use $\mathcal{G}_{sub}$ to denote a subgraph of $\mathcal{G}$, and use $\bar{\mathcal{G}}_{sub}$ to denote the complementary structure of $\mathcal{G}_{sub}$ in $\mathcal{G}$. We also introduce the prototype layer, which consists of a set of prototypes $\mathcal{Z}_p = \left\{ \mathbf{z}_{\mathcal{G}_p}^1, \mathbf{z}_{\mathcal{G}_p}^2, \cdots, \mathbf{z}_{\mathcal{G}_p}^M \right\}$, where $M$ is the total number of prototypes, and each prototype $\mathbf{z}_{\mathcal{G}_p}^m$ is a learnable parameter vector that serves as the latent representation of the prototypical part (i.e., $\mathcal{G}_p$) of graph $\mathcal{G}$. We allocate $J$ prototypes for each class, i.e., $M = K \times J$.

**Graph Information Bottleneck.** The mutual information between two random variables $X$ and $Y$, i.e., $I(X; Y)$, is defined as follows:

$$I(X; Y) = \int_X \int_Y p(x, y) \log \frac{p(x, y)}{p(x)p(y)} \mathrm{d}x \mathrm{d}y \tag{1}$$

Given the input $X$ and its associated label $Y$, the Information Bottleneck (IB) [21] aims to obtain a bottleneck variable $Z$ by optimizing the following objective:

$$\min_Z -I(Y; Z) + \beta I(X; Z), \tag{2}$$

where $\beta$ is the Lagrange multiplier used to control the trade-off between the two terms. IB principle has recently been applied to learning a bottleneck graph, named IB-Graph, for a given graph $\mathcal{G}$, which retains the minimal sufficient information in terms of $\mathcal{G}$'s properties [31]. This approach is motivated by the Graph Information Bottleneck (GIB) principle, which seeks to identify an informative yet compressed subgraph $\mathcal{G}_{sub}$ from the original graph $\mathcal{G}$ by optimizing the following objective:

$$\min_{\mathcal{G}_{sub}} -I(Y; \mathcal{G}_{sub}) + \beta I(\mathcal{G}; \mathcal{G}_{sub}), \tag{3}$$

where $Y$ is the label of $\mathcal{G}$. The first term maximizes the mutual information between the graph label and the compressed subgraph, which ensures that the compressed subgraph contains as much information as possible about the graph label. The second term minimizes the mutual information between the input graph and the compressed subgraph, which ensures that the compressed subgraph contains minimal information about the input graph.

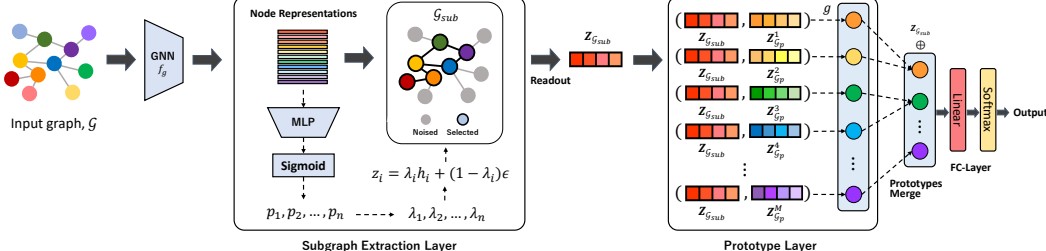

Figure 2: The architecture of our proposed PGIB. PGIB generates a subgraph $\mathcal{G}_{sub}$ by injecting noise to identify core subgraphs, and it is used to compute similarity scores between prototypes in the prototype layer. The trained prototypes play a crucial role in visualizing the reasoning processes during training in an interpretable manner. PGIB also involves merging pairs of similar prototypes to decrease the number of prototypes. Finally, the integrated prototypes are utilized to predict the graph labels in the fully connected layer.

# 3 Methodology

In this section, we present our proposed method, called PGIB. We introduce Prototype-based Graph Information Bottleneck (Section 3.1), each layer in the architecture (Section 3.2 - 3.4), and the interpretability stabilization process (Section 3.5), which enhances the interpretability and the tracking capabilities of the reasoning process during the model training.

**Model Architecture.** Figure 2 presents an overview of PGIB. We first generate the representations of nodes in the input graph $\mathcal{G}$ using a GNN encoder. Then, the node representations are passed to the subgraph extraction layer that assigns each node in $\mathcal{G}$ to either $\mathcal{G}_{sub}$ or $\bar{\mathcal{G}}_{sub}$. Next, we compute the similarities between the embedding $\mathbf{z}_{\mathcal{G}_{sub}}$ and the set of prototypes $\mathcal{Z}_p = \left\{ \mathbf{z}_{\mathcal{G}_p}^1, \mathbf{z}_{\mathcal{G}_p}^2, \cdots, \mathbf{z}_{\mathcal{G}_p}^M \right\}$ in the prototype layer. Finally, we merge the prototypes that are semantically similar, which are then used to generate the final prediction.

## 3.1 Prototype-based Graph Information Bottleneck

PGIB is a novel explainable GNN framework that incorporates the prototype learning within the IB framework, thereby enabling the prototypes to capture the essential key subgraph of the input graph detected by the IB framework. More precisely, we reformulate the GIB objective shown in Equation 3 by decomposing the first term, i.e., $I(Y; \mathcal{G}_{sub})$, with respect to the prototype $\mathcal{G}_p$ using the chain rule of mutual information in order to examine the impact of the joint information between $\mathcal{G}_{sub}$ and $\mathcal{G}_p$ on $Y$ as follows:

$$\min_{\mathcal{G}_{sub}} \underbrace{-I(Y; \mathcal{G}_{sub}, \mathcal{G}_p) + I(Y; \mathcal{G}_p | \mathcal{G}_{sub})}_{\text{Section 3.3}} + \beta \underbrace{I(\mathcal{G}; \mathcal{G}_{sub})}_{\text{Section 3.2}} . \tag{4}$$

Please refer to Appendix A.1 for a detailed proof of Equation 4. In the following sections, we describe how each term is optimized during training.

## 3.2 Subgraph Extraction Layer (Minimizing $I(\mathcal{G}; \mathcal{G}_{sub})$)

The goal of the subgraph extraction layer is to extract an informative subgraph $\mathcal{G}_{sub}$ from $\mathcal{G}$ that contains minimal information about $\mathcal{G}$. We minimize $I(\mathcal{G}; \mathcal{G}_{sub})$ by training the model to inject noise into insignificant subgraphs $\bar{\mathcal{G}}_{sub}$, while injecting less noise into more informative ones $\mathcal{G}_{sub}$ [32]. Specifically, given the representation of node $v_i$, i.e., $\mathbf{h}_i$, we compute the probability $p_i$ with an MLP followed by a sigmoid function, which is then used to replace the representation $h_i$ to obtain the final representation $\mathbf{z}_i$ as follows:

$$p_i = \text{Sigmoid}(\text{MLP}(h_i))$$
$$z_i = \lambda_i h_i + (1 - \lambda_i)\epsilon, \quad \text{where } \lambda_i \sim \text{Bernoulli}(p_i) \text{ and } \epsilon \sim \mathcal{N}(\mu_{\mathbf{h}_i}, \sigma_{\mathbf{h}_i}^2). \tag{5}$$

That is, the learned probability $p_i$ enables selective preservation of information in $\mathcal{G}_{sub}$, and based on this probability, the quantity of information transmitted from $\mathbf{h}_i$ to $\mathbf{z}_i$ can be flexibly adjusted to compress the information from $\mathcal{G}$ to $\mathcal{G}_{sub}$. This approach not only retains interpretability within the

subgraph itself, but also potentially facilitates the learning of prototypes that are introduced in the next step. Following [32], we minimize the upper bound of $I(\mathcal{G}; \mathcal{G}_{sub})$ as follows:

$$I(\mathcal{G}; \mathcal{G}_{sub}) \leq \mathbb{E}_{\mathcal{G}}(-\frac{1}{2}\log A + \frac{1}{2|\mathcal{V}_{\mathcal{G}}|}A + \frac{1}{2|\mathcal{V}_{\mathcal{G}}|}B^2) =: \mathcal{L}_{\mathrm{MI}}^1(\mathcal{G}, \mathcal{G}_{sub}), \quad (6)$$

where $A = \sum_{i=1}^{|\mathcal{V}_{\mathcal{G}}|}(1 - \lambda_i)^2$ and $B = \frac{\sum_{i=1}^{|\mathcal{V}_{\mathcal{G}}|} \lambda_i(\mathbf{h}_i - \mu_{\mathbf{h}_i})}{\sigma_{\mathbf{h}_i}}$. Thus, minimizing $\mathcal{L}_{\mathrm{MI}}^1$ allows us to minimize the upper bound of $I(\mathcal{G}; \mathcal{G}_{sub})$. After noise injection, we compute the embedding $\mathbf{z}_{\mathcal{G}_{sub}}$ through a graph readout function such as max pooling and sum pooling. For further details and analysis of the different graph readout functions, please refer to Appendix A.4.2.

### 3.3 Prototype Layer (Minimizing $-I(Y; \mathcal{G}_{sub}, \mathcal{G}_p) + I(Y; \mathcal{G}_p|\mathcal{G}_{sub})$ )

The prototype layer involves allocation of a fixed number of prototypes for each class. The prototypes are required to capture the most significant graph patterns that can aid in the identification of the graphs within each class. To begin with, we define the similarity score between the prototype $\mathbf{z}_{\mathcal{G}_p}$ and the embedding $\mathbf{z}_{\mathcal{G}_{sub}}$ obtained from noise injection as follows:

$$g(\mathbf{z}_{\mathcal{G}_{sub}}, \mathbf{z}_{\mathcal{G}_p}) = \log\left(\frac{\|\mathbf{z}_{\mathcal{G}_{sub}} - \mathbf{z}_{\mathcal{G}_p}\|_2^2 + 1}{\|\mathbf{z}_{\mathcal{G}_{sub}} - \mathbf{z}_{\mathcal{G}_p}\|_2^2 + \epsilon}\right), \quad (7)$$

where $\mathbf{z}_{\mathcal{G}_p}$ is the prototype and shares the same dimension as $\mathbf{z}_{\mathcal{G}_{sub}}$.

#### 3.3.1 Minimizing $-I(Y; \mathcal{G}_{sub}, \mathcal{G}_p)$

We derive the lower bound of $I(Y; \mathcal{G}_{sub}, \mathcal{G}_p)$ as follows:

*Proposition* 1. (**Lower bound of** $I(Y; \mathcal{G}_{sub}, \mathcal{G}_p)$) Given significant subgraph $\mathcal{G}_{sub}$ for a graph $\mathcal{G}$, its label information $Y$, prototype graph $\mathcal{G}_p$ and similarity function $\gamma$, we have

$$\begin{aligned}
I(Y; \mathcal{G}_{sub}, \mathcal{G}_p) &= \mathbb{E}_{Y, \mathcal{G}_{sub}, \mathcal{G}_p}\left[\log p\left(Y|\mathcal{G}_{sub}, \mathcal{G}_p\right)\right] - \mathbb{E}_Y\left[\log p(Y)\right] \\
&\geq \mathbb{E}_{Y, \mathcal{G}_{sub}, \mathcal{G}_p}\left[\log p\left(Y|\gamma\left(\mathcal{G}_{sub}, \mathcal{G}_p\right)\right)\right] - \mathbb{E}_Y\left[\log p(Y)\right] \\
&\geq \mathbb{E}_{Y, \mathcal{G}_{sub}, \mathcal{G}_p}\left[\log q_\theta\left(Y|\gamma\left(\mathcal{G}_{sub}, \mathcal{G}_p\right)\right)\right] \\
&=: -\mathcal{L}_{\mathrm{cls}}(q_\theta\left(Y|\gamma\left(\mathcal{G}_{sub}, \mathcal{G}_p\right)\right))
\end{aligned} \quad (8)$$

where $q_\theta\left(Y|\gamma\left(\mathcal{G}_{sub}, \mathcal{G}_p\right)\right)$ is the variational approximation to the true posterior $p\left(Y|\gamma\left(\mathcal{G}_{sub}, \mathcal{G}_p\right)\right)$.

Equation 8 demonstrates that the maximization of the mutual information $I(Y; \mathcal{G}_{sub}, \mathcal{G}_p)$ can be attained by minimizing the classification loss, denoted as $\mathcal{L}_{cls}$. This maximization of mutual information between the label $Y$ and the similarity information $\gamma\left(\mathcal{G}_{sub}, \mathcal{G}_p\right)$ promotes the subgraph $\mathcal{G}_{sub}$ and prototype $\mathcal{G}_p$ to possess predictive capabilities concerning the graph label $Y$. In practical applications, the cross-entropy loss is chosen for a categorical $Y$. For a comprehensive understanding of the derivation process of Equation 8, refer to the Appendix A.2. Note that the similarity between $\mathcal{G}_{sub}$ and $\mathcal{G}_p$, i.e., $\gamma\left(\mathcal{G}_{sub}, \mathcal{G}_p\right)$, is computed by the similarity score defined in Equation 7, i.e., $g(\mathbf{z}_{\mathcal{G}_{sub}}, \mathbf{z}_{\mathcal{G}_p})$.

#### 3.3.2 Minimizing $I(Y; \mathcal{G}_p|\mathcal{G}_{sub})$

We investigate the mutual information, denoted as $I(Y; \mathcal{G}_p|\mathcal{G}_{sub})$, from the perspective of the interaction between $\mathcal{G}_{sub}$ and $\mathcal{G}_p$. We decompose $I(Y; \mathcal{G}_p|\mathcal{G}_{sub})$ into the sum of two terms based on the chain rule of mutual information as follows:

$$I(Y; \mathcal{G}_p|\mathcal{G}_{sub}) = I(\mathcal{G}_p; Y, \mathcal{G}_{sub}) - I(\mathcal{G}_{sub}; \mathcal{G}_p). \quad (9)$$

It is important to note that the first term, i.e., $I(\mathcal{G}_p; Y, \mathcal{G}_{sub})$, minimizes the mutual information between $\mathcal{G}_p$ and the joint variables $(Y, \mathcal{G}_{sub})$, which eliminates the information about $Y$ related to $\mathcal{G}_{sub}$ from $\mathcal{G}_p$. However, since our goal is not to solely minimize $I(\mathcal{G}_p; Y, \mathcal{G}_{sub})$ but to ensure the interpretability of the prototype $\mathcal{G}_p$, including this term leads to diminished interpretability of $\mathcal{G}_p$. Consequently, we excluded the first term during training, and only consider the second term, i.e., $-I(\mathcal{G}_{sub}; \mathcal{G}_p)$, to simultaneously guarantee the interpretability of both $\mathcal{G}_{sub}$ and $\mathcal{G}_p$. A detailed derivation for Equation 9 is given in Appendix A.3. From now on, we describe approaches for minimizing the second term. Inspired by [11], we introduce two different approaches for minimizing $-I(\mathcal{G}_{sub}; \mathcal{G}_p)$.

**1) Variational IB-based approach.** We obtain the upper bound of $-I(\mathcal{G}_{sub}; \mathcal{G}_p)$ using the variational IB-based approach as follows:

$$-I(\mathcal{G}_{sub}; \mathcal{G}_p) \leq \mathbb{E}_{\mathcal{G}_{sub}, \mathcal{G}_p}[-\log q_\phi(\mathcal{G}_p | \mathcal{G}_{sub})] := \mathcal{L}_{\mathrm{MI}}^2(\mathcal{G}_{sub}, \mathcal{G}_p), \quad (10)$$

where $q_\phi(\mathcal{G}_p | \mathcal{G}_{sub})$ is the variation approximation of $p(\mathcal{G}_p | \mathcal{G}_{sub})$. Equation 10 shows that the maximization of the mutual information $I(\mathcal{G}_{sub}; \mathcal{G}_p)$ can be attained by minimizing $\mathcal{L}_{\mathrm{MI}}^2(\mathcal{G}_{sub}, \mathcal{G}_p)$. We select a single-layer linear transformation as a modeling option for $q_\phi$ to minimize the information loss of $\mathcal{G}_{sub}$ when predicting $\mathcal{G}_p$.

**2) Contrastive learning-based approach.** Recent studies on contrastive learning [20, 30, 7] have proven that minimizing contrastive loss is equivalent to maximizing the mutual information between two variables. Hence, we additionally propose a variant of PGIB, i.e., PGIB$_{\mathsf{cont}}$, that minimizes the contrastive loss instead of minimizing the lower bound as defined in Equation 10. More precisely, we consider $\mathcal{G}_p$ and $\mathcal{G}_{sub}$ with the same label as a positive pair, and the contrastive loss is defined as follows:

$$\mathcal{L}_{\mathrm{MI}}^2 = -\frac{1}{n} \sum_{i=1}^{n} \log \frac{\sum_{j:\mathbf{z}_{\mathcal{G}_p^j} \in \mathbb{P}_{y_i}} \exp(g(\mathbf{z}_{\mathcal{G}_{sub}^i}, \mathbf{z}_{\mathcal{G}_p^j})/\tau)}{\sum_{k:\mathbf{z}_{\mathcal{G}_p^k} \notin \mathbb{P}_{y_i}} \exp(g(\mathbf{z}_{\mathcal{G}_{sub}^i}, \mathbf{z}_{\mathcal{G}_p^k})/\tau)}. \quad (11)$$

where $\tau$ is the temperature hyperparameter, $n$ denotes the number of graphs in a batch, and $j$ and $k$ indicate indices of positive and negative samples, respectively. $\mathbb{P}_{y_i}$ is the set of prototypes that belong to class $y_i$. We effectively confer interpretability to $\mathcal{G}_p$ by increasing its similarity with each $\mathcal{G}_{sub}$.

### 3.4 Prediction Layer

We obtain the set of similarity scores $\mathbf{r} \in \mathbb{R}^M$, whose $m$-th element $r_m = g(\mathbf{z}_{\mathcal{G}_{sub}}, \mathbf{z}_{\mathcal{G}_p}^m)$ denotes the similarity score between $\mathbf{z}_{\mathcal{G}_{sub}}$ and $\mathbf{z}_{\mathcal{G}_p}$ as defined in Equation 7. Then, we compute the final predicted probability $\boldsymbol{\pi} \in \mathbb{R}^K$ by passing $\mathbf{r}$ and $\mathbf{z}_{\mathcal{G}_{sub}}$ through a linear layer with weights $\omega$, followed by the softmax function. Specifically, $\omega[m, :]$ denotes weight assigned to $r_m$, i.e., $\omega(\mathbf{z}_{\mathcal{G}_p}^m)$. Finally, we calculate the cross-entropy classification loss, as follows:

$$\mathcal{L}_{\mathrm{cls}} = -\frac{1}{N} \sum_{i=1}^{N} \sum_{c=1}^{K} \mathbb{I}(y_i = c) \log(\pi_c). \quad (12)$$

### 3.5 Interpretability Stabilization

**Merging Prototypes.** Since the number of prototypes for each class is determined before training, some of the learned prototypes may share similar semantics, which negatively affects the model interpretability for which the small size and low complexity are desirable [6, 25]. Inspired by [17], we propose a method to effectively merge the prototypes for graph-structured data, which, in turn, enhances the explanation of the reasoning process and improves performance on downstream tasks while reducing model complexity. The main idea is to merge prototypes based on the similarity between prototype pairs using the embeddings $\mathbf{z}_{\mathcal{G}sub}$. This similarity utilizes all training subgraphs (i.e., $\bigcup_{\mathcal{G} \in \mathcal{X}} \mathcal{G}_{sub}$, where $\mathcal{X}$ is the training set) by measuring the disparity between $g(\mathbf{z}_{\mathcal{G}_{sub}}, \mathbf{z}_{\mathcal{G}_p^i})$ and $g(\mathbf{z}_{\mathcal{G}_{sub}}, \mathbf{z}_{\mathcal{G}_p^j})$ as follows:

$$h(\mathbf{z}_{\mathcal{G}_p^i}, \mathbf{z}_{\mathcal{G}_p^j}) = \left[ \sum_{\mathcal{G} \in \mathcal{X}} (g(\mathbf{z}_{\mathcal{G}_{sub}}, \mathbf{z}_{\mathcal{G}_p^i}) - g(\mathbf{z}_{\mathcal{G}_{sub}}, \mathbf{z}_{\mathcal{G}_p^j}))^2 \right]^{-1}. \quad (13)$$

Then, for every pair $(\mathbf{z}_{\mathcal{G}_p^i}, \mathbf{z}_{\mathcal{G}_p^j})$ that falls within the highest $\xi$ percent of similar pairs, the prototype $\mathbf{z}_{\mathcal{G}_p^j}$ and its corresponding weights $\omega(\mathbf{z}_{\mathcal{G}_p^j})$ are removed, and the weights $\omega(\mathbf{z}_{\mathcal{G}_p^i})$ are updated to the sum of $\omega(\mathbf{z}_{\mathcal{G}_p^i})$ and $\omega(\mathbf{z}_{\mathcal{G}_p^j})$. We combine the $\xi$ percentage of the most similar prototype pairs based on the calculated similarity scores.

**Prototype Projection.** Since the learned prototypes are embedding vectors that cannot be directly interpreted, we project each prototype $\mathbf{z}_{\mathcal{G}_p}$ onto the nearest latent training subgraph from the same class. This process establishes a conceptual equivalence between each prototype and a training subgraph, thereby enhancing interpretability of the prototypes. Specifically, we update prototype $\mathbf{z}_{\mathcal{G}_p}$ of class $k$ ($i.e., \mathbf{z}_{\mathcal{G}_p} \in \mathbb{P}_k$) by performing the following operation:

$$\mathbf{z}_{\mathcal{G}_p} \leftarrow \underset{\tilde{\mathbf{z}} \in \mathbb{Z}}{\arg\min} \|\tilde{\mathbf{z}} - \mathbf{z}_{\mathcal{G}_p}\|_2, \text{ where } \mathbb{Z} = \left\{\tilde{\mathbf{z}} : \text{Readout}\{f_g(\tilde{\mathcal{G}})\}, \tilde{\mathcal{G}} \in \text{Subgraph}(\mathcal{G}_i) \; \forall i \; s.t. \; y_i = k\right\}. \quad (14)$$

In the Equation 14, we use Monte Carlo Tree Search (MCTS) [18] to explore training subgraphs $\tilde{\mathcal{G}}$ during prototype projection.

**Connectivity Loss.** For an input graph $\mathcal{G}$, we construct a node assignment $S_{\mathcal{G}} \in \mathbb{R}^{|\mathcal{V}_{\mathcal{G}}| \times 2}$ based on the probability values that are computed by Equation 5. Specifically, $S_{\mathcal{G}}[j, 0]$ and $S_{\mathcal{G}}[j, 1]$ denote the probability of node $v_j \in \mathcal{V}_{\mathcal{G}}$ belonging to $\mathcal{G}_{sub}$ and $\bar{\mathcal{G}}_{sub}$, respectively. Following [31], poor initialization of the matrix $S$ may result in the proximity of its elements $S[j, 0]$ and $S[j, 1]$ for $\forall v_j \in \mathcal{V}_{\mathcal{G}_i}$, leading to an unstable connectivity of $\mathcal{G}_{sub}$. This instability can have adverse effects on the subgraph generation process. To enhance the interpretability of $\mathcal{G}_{sub}$ by inducing a compact topology, we utilize a batch-wise loss function as follows:

$$\mathcal{L}_{\text{con}} = \|\text{Norm}(S_{\text{B}}^T \mathbf{A}_{\text{B}} S_{\text{B}}) - I_2\|_F \quad (15)$$

where $S_{\text{B}} \in \mathbb{R}^{\sum_{i=1}^{n} |\mathcal{V}_{\mathcal{G}_i}| \times 2}$ and $\mathbf{A}_{\text{B}} \in \mathbb{R}^{\sum_{i=1}^{n} |\mathcal{V}_{\mathcal{G}_i}| \times \sum_{i=1}^{n} |\mathcal{V}_{\mathcal{G}_i}|}$ are the node assignment and the adjacency matrix at the batch level, respectively. $I_2$ is 2-by-2 identity matrix, $\| \cdot \|_F$ is the Frobenius norm and $\text{Norm}(\cdot)$ is the row normalization. Minimizing $\mathcal{L}_{\text{con}}$ indicates that if $v_j$ is in $\mathcal{G}_{sub}$ its neighbors also have a high probability to be in $\mathcal{G}_{sub}$, while if $v_i$ is in $\mathcal{G}_{sub}$, its neighbors have a low probability to be in $\bar{\mathcal{G}}_{sub}$.

**Final Objectives**. Finally, we define the objective of our model as the sum of the losses as follows:

$$\mathcal{L}_{\text{total}} = \mathcal{L}_{\text{cls}} + \alpha_1 \mathcal{L}_{\text{MI}}^1 + \alpha_2 \mathcal{L}_{\text{MI}}^2 + \alpha_3 \mathcal{L}_{\text{con}} \quad (16)$$

where $\alpha_1, \alpha_2$ and $\alpha_3$ are hyper-parameters that adjust the weights of the losses. A detailed ablation study for each loss term is provided in Appendix A.4.1 for further analysis.

## 4 Experiments

### 4.1 Experimental Settings

Each dataset is split into training, validation, and test sets with a ratio of $80\%$, $10\%$, and $10\%$, respectively. All models are trained for 300 epochs using the Adam optimizer with a learning rate of 0.005. GIN [28] is used as the encoder for all models used in the experiment. We evaluate the performance based on accuracy, which is averaged over 10 independent runs with different random seeds. For simplicity, the hyperparameters $\alpha_1, \alpha_2$, and $\alpha_3$ in Equation 16 are set to $0.0001, 0.01$ to $0.1$ and $5$, respectively. The prototype merge operation starts at epoch 100 and is performed every 50 epochs thereafter. We set the number of prototypes per class to 7 and combine $30\%$ of the most similar prototype pairs.

### 4.2 Graph Classification

**Datasets and Baselines**. We use the MUTAG [16], PROTEINS [1], NCI1 [24], and DD[5] datasets. These are datasets related to molecules or bioinformatics, and are widely used for evaluations on graph classification. We consider three GNN baselines, including GCN [10], GIN [28], GAT [22]. In addition, we compare PGIB with several state-of-the-art built-in models that integrate explanation functionality internally, including a prototype-based method ProtGNN [35], and IB-based models such as GIB [31], VGIB [32], and GSAT [13]. Further details about the baselines and datasets are provided in Appendix A.5 and A.6, respectively.

**Experiment Results**. Experimental results for graph classification are presented in the Table 1. In the table, PGIB and PGIB$_{\text{cont}}$ represent our proposed methods. PGIB utilizes a Variational IB-based approach, while PGIB$_{\text{cont}}$ employs a Contrastive learning-based approach to maximize $I(\mathcal{G}_{\text{sub}}; \mathcal{G}_p)$ (Section 3.3.2). We have the following observations: **1)** All variants of PGIB outperform the baselines including both the prototype-based and IB-based methods on all datasets. Notably, PGIBs incorporate the crucial information of the key subgraph, which significantly contributes to enhancing the classification performance. PGIB$_{\text{cont}}$ achieves a significant improvement of up to $5.6\%$ compared to the runner-up baseline. **2)** We observe that PGIB$_{\text{cont}}$ performs relatively better than PGIB. We attribute this to the nature of the contrastive loss, which is generally shown to be effective in classifying instances between different classes, allowing the prototypes learned based on the contrastive loss to be more distinguishable from one another.

Table 1: Evaluation on graph classification (accuracy).

| Dataset | Methods | | | | | | | | |
|---------|---------|-----|-----|---------|-----|------|------|------|------|
| | GCN | GIN | GAT | ProtGNN | GIB | VGIB | GSAT | **PGIB** | **PGIB$_{cont}$** |
| MUTAG | 74.50±7.89 | 80.50±7.89 | 73.50±7.43 | 80.50±9.07 | 79.00±6.24 | 81.00±6.63 | 80.88±8.94 | 85.00±7.07 | **85.50±5.22** |
| PROTEINS | 72.83±4.23 | 70.30±4.84 | 71.35±4.85 | 73.83±4.22 | 75.25±5.92 | 73.66±3.32 | 69.64±4.71 | 77.14±2.19 | **77.50±2.42** |
| NCI1 | 73.16±3.49 | 75.04±2.08 | 66.05±1.03 | 74.13±2.10 | 64.65±6.78 | 63.75±3.37 | 68.13±2.64 | 77.65±2.20 | **78.25±2.13** |
| DD | 72.53±4.51 | 72.04±3.62 | 70.81±4.33 | 69.15±4.33 | 72.61±8.26 | 72.77±5.63 | 71.93±2.74 | 73.36±1.80 | **73.70±2.14** |

## 4.3 Graph Interpretation

In this section, we evaluate the process of extracting subgraphs that possess the most similar properties of the original graph. We present qualitative results including subgraph visualizations, and conduct quantitative experiments to measure how accurately explanations capture the important components that contribute to the model's predictions.

**Datasets and Baselines**. We use four molecular properties from the ZINC [8] dataset, which consists of 250,000 molecules, for graph interpretation. QED measures the likelihood of a molecule being a drug and DRD2 indicates the probability of a molecule being active on dopamine type 2 receptors. HLM-CLint and RLM represent estimates of in vitro human and rat liver microsomal metabolic stability (mL/min/g as base 10 logarithm). We compare PGIB$_{cont}$ with several representative interpretation models, including GNNexplainer [29], PGexplainer [12], GIB [31], and VGIB [32]. Note that as PGIB shows similar results with PGIB$_{cont}$, we only report the results of PGIB$_{cont}$ for simplicity. Further details on baselines and datasets are described in Appendix A.5 and A.6, respectively.

**Qualitative Analysis**. Figure 3(a)-(d) present the visualization of subgraphs in Mutag dataset. According to [29, 4], the $NO_2$ functional group is known to be a cause of mutagenicity, while the carbon ring is a substructure that is not related to mutagenicity. In the figure, the bold edges connect the nodes that the models consider important. The $NO_2$ group in Mutag is correctly identified by PGIB, while VGIB, ProtGNN, and GNNexplainer fail to recognize all $NO_2$ groups or include other unnecessary substructures. Figure 3(e)-(h) present the visualization of subgraphs in BA-2Motifs dataset. We observe that PGIB accurately recognizes motif graphs containing the label information such as a house or a five-node cycle, but other models have difficulty in detecting the complete motifs.

**Quantitative Analysis**. Although visualizing the explanations generated by models plays a crucial role in assessing various explanation models, relying solely on qualitative evaluations may not always be reliable due to their subjective nature. Therefore, we also perform quantitative experiments using the Fidelity metric [14, 34].

The Fidelity metric quantifies the extent to which explanations accurately capture the important components that contribute to the model's predictions. Specifically, let $y_i$ and $\hat{y}_i$ denote the ground-truth and predicted values for the $i$-th input graph, respectively. Moreover, $k$ denotes the sparsity score of the selected subgraph in which nodes whose importance scores obtained by Equation 5 are among the top-$(k \times 100)\%$ within the original graph, and its prediction is denoted by $\hat{y}_i^k$. Additionally, $\hat{y}_i^{1-k}$ denotes the prediction based on the remaining subgraph. The Fidelity scores are computed as follows:

$$\mathcal{F}_- = \frac{1}{N} \sum_{i=1}^{N} \mathbb{I}(y_i = \hat{y}_i) - \mathbb{I}(y_i = \hat{y}_i^k), \quad \mathcal{F}_+ = \frac{1}{N} \sum_{i=1}^{N} \mathbb{I}(y_i = \hat{y}_i) - \mathbb{I}(y_i = \hat{y}_i^{1-k}), \quad (17)$$

where $\mathbb{I}(y_i = \hat{y}_i)$ is the binary indicator which returns 1 if $y_i = \hat{y}_i$, and 0 otherwise. In other words, they measure how well the predictions made solely based on the extracted subgraph (i.e., $\mathcal{F}_-$) and the remaining subgraph (i.e., $\mathcal{F}_+$) mimic the predictions made based on the entire graph, respectively. Hence, a low value of $\mathcal{F}_-$ and a high value of $\mathcal{F}_-$ indicate better explainability of the model.

Table 2 shows the fidelity scores on four datasets at the sparsity score of $k = 0.5$. Our proposed model outperforms both post-hoc and built-in state-of-the-art explanation models in all datasets. Furthermore, merging prototypes achieves significant improvements in terms of interpretability. This implies

Table 2: Evaluation on graph interpretation (Fidelity scores).

| Method | $\mathcal{F}_- \downarrow$ | | | | $\mathcal{F}_+ \uparrow$ | | | |
|--------|-----|----------|-----|------|-----|----------|-----|------|
| | RLM | HLM-CLint | QED | DRD2 | RLM | HLM-CLint | QED | DRD2 |
| GNNexplainer | 0.478 | 0.616 | 0.498 | 0.433 | 0.694 | 0.778 | 0.602 | 0.740 |
| PGexplainer | 0.502 | 0.620 | 0.560 | 0.540 | 0.632 | 0.692 | 0.598 | 0.686 |
| GIB | 0.483 | 0.643 | 0.525 | 0.428 | 0.654 | 0.781 | 0.601 | 0.724 |
| VGIB | 0.463 | 0.579 | 0.487 | 0.424 | 0.765 | 0.792 | 0.627 | 0.756 |
| PGIB$_{cont}$ | 0.441 | 0.593 | 0.459 | 0.406 | 0.747 | 0.772 | 0.613 | 0.771 |
| PGIB$_{cont}$ + merge | **0.415** | **0.543** | **0.447** | **0.379** | **0.765** | **0.796** | **0.635** | **0.781** |

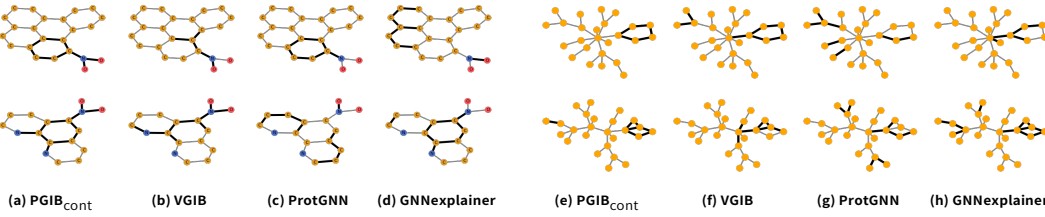

| (a) PGIB$_{cont}$ | (b) VGIB | (c) ProtGNN | (d) GNNexplainer | (e) PGIB$_{cont}$ | (f) VGIB | (g) ProtGNN | (h) GNNexplainer |

Figure 3: Explanation visualizations on Mutag (a-d) and BA-2Motifs (e-h)

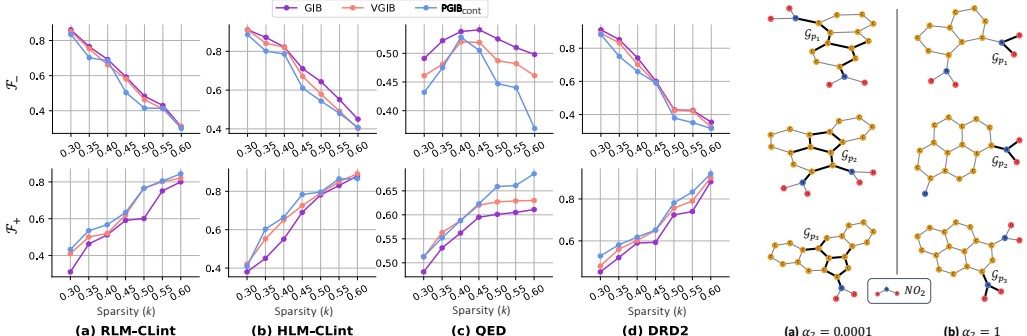

| (a) RLM-CLint | (b) HLM-CLint | (c) QED | (d) DRD2 | (a) $\alpha_2 = 0.0001$ | (b) $\alpha_2 = 1$ |

Figure 4: Comparisons of fidelity scores over sparsity scores $k$.  Figure 5: $\mathcal{G}_p$ visualization over $\alpha_2$.

that decreasing the number of prototypes can eliminate uninformative substructures and emphasize key substructures, which increases the interpretability of the extracted subgraphs. Figure 4 visualizes the comparison of fidelity scores over various sparsity scores of subgraphs. To ensure a fair comparison, the fidelity scores are compared under the same subgraph sparsity, as the difference between the predictions of the original graph and subgraph strongly depends on the level of sparsity. We observe that PGIB$_{cont}$ achieves the best performance in most sparsity environments on the four datasets.

### 4.4 Hyperparameter Analysis

In Figure 6, we conduct a sensitivity analysis on the hyperparameters $\alpha_1$ and $\alpha_2$ of the final loss (Equation 16) relevant to mutual information. Note that $\alpha_1$ and $\alpha_2$ are related to minimizing $I(\mathcal{G}; \mathcal{G}_{sub})$ and maximizing $I(\mathcal{G}_{sub}; \mathcal{G}_p)$, respectively. **1)** Figure 6(a) shows a significant decrease in performance when $\alpha_1$ becomes large, i.e., when the model focuses on compressing the subgraphs. This is because too much compression of

Figure 6: Impact of $\alpha_1$ and $\alpha_2$ on PROTEINS dataset.

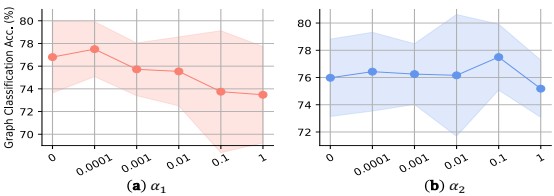

| (a) $\alpha_1$ | (b) $\alpha_2$ |

subgraphs results in the loss of important information, ultimately having a negative impact on the downstream performance. However, when $\alpha_1 = 0$ (i.e., $\mathcal{G} = \mathcal{G}_{sub}$; no compression at all), uninformative information would be included in $\mathcal{G}_{sub}$, which incurs a performance degradation. **2)** Figure 6(b) visualizes the change in performance depending on $\alpha_2$. A small value of $\alpha_2$ prevents sufficient transmission of information from $\mathcal{G}_{sub}$ to $\mathcal{G}_p$, whereas excessive value of $\alpha_2$ allows the influence of $\mathcal{G}_{sub}$ to dominate the prototypes $\mathcal{G}_p$, both of which lead to a performance deterioration. For example, in Mutag dataset, a low value of $\alpha_2$ ultimately results in $\mathcal{G}_p$ not obtaining label-relevant information (i.e., $NO_2$) that is captured by $\mathcal{G}_{sub}$ (See Figure 5(a)). On the other hand, a high value of $\alpha_2$ hinders the formation of diverse prototypes (See Figure 5(b)).

## 5  Conclusion and Future work

We propose a novel framework of explainable GNNs, called interpretable Prototype-based Graph Information Bottleneck (PGIB), that integrates prototype learning into the information bottleneck framework. The main idea of PGIB is to learn prototypes that capture subgraphs containing key structures relevant to the label information, and to merge the semantically similar prototypes for better model interpretability and model complexity. Experimental results show that PGIB achieves improvements not only in the performance on downstream tasks, but also provides more precise

explanation of the reasoning process. For future work, we plan to further extend the applicability of PGIB by integrating domain knowledge into prototype learning by imposing constraints on subgraphs. We expect that this approach would enable the learning of prototypes that align with the domain knowledge as they obtain domain-specific information from the subgraphs.

## Acknowledgments and Disclosure of Funding

This work was supported by Institute of Information & Communications Technology Planning & Evaluation(IITP) grant funded by the Korea government(MSIT) (RS-2023-00216011, No.2022-0-00077), and the National Research Foundation of Korea(NRF) funded by Ministry of Science and ICT (NRF-2022M3J6A1063021).

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
