# A    Appendix

## A.1    Prototype-based Graph Information Bottleneck - Eq. 4

From Eq. 3, the GIB objective is :

$$\min_{\mathcal{G}_{sub}} -I(Y;\mathcal{G}_{sub}) + \beta I(\mathcal{G};\mathcal{G}_{sub}). \tag{18}$$

For the first term $-I(Y;\mathcal{G}_{sub})$, by definition:

$$-I(Y;\mathcal{G}_{sub}) = -\mathbb{E}_{Y,\mathcal{G}_{sub}} \left[ \log \frac{p(Y,\mathcal{G}_{sub})}{p(Y)p(\mathcal{G}_{sub})} \right]. \tag{19}$$

We allow the involvement of $\mathcal{G}_p$ in Eq. 19 as follows:

$$
\begin{aligned}
-I(Y;\mathcal{G}_{sub}) &= -\mathbb{E}_{Y,\mathcal{G}_{sub},\mathcal{G}_p} \left[ \log \frac{p(Y,\mathcal{G}_{sub},\mathcal{G}_p)}{p(\mathcal{G}_{sub},\mathcal{G}_p)p(Y)} \right] - \mathbb{E}_{Y,\mathcal{G}_{sub},\mathcal{G}_p} \left[ \log \frac{p(Y,\mathcal{G}_{sub})p(\mathcal{G}_{sub},\mathcal{G}_p)}{p(Y,\mathcal{G}_{sub},\mathcal{G}_p)p(\mathcal{G}_{sub})} \right] \\
&= -\mathbb{E}_{Y,\mathcal{G}_{sub},\mathcal{G}_p} \left[ \log \frac{p(Y,\mathcal{G}_{sub},\mathcal{G}_p)}{p(\mathcal{G}_{sub},\mathcal{G}_p)p(Y)} \right] + \mathbb{E}_{Y,\mathcal{G}_{sub},\mathcal{G}_p} \left[ \log \frac{p(Y,\mathcal{G}_{sub},\mathcal{G}_p)p(\mathcal{G}_{sub})}{p(Y,\mathcal{G}_{sub})p(\mathcal{G}_{sub},\mathcal{G}_p)} \right] \\
&= -\mathbb{E}_{Y,\mathcal{G}_{sub},\mathcal{G}_p} \left[ \log \frac{p(Y,\mathcal{G}_{sub},\mathcal{G}_p)}{p(\mathcal{G}_{sub},\mathcal{G}_p)p(Y)} \right] + \mathbb{E}_{Y,\mathcal{G}_{sub},\mathcal{G}_p} \left[ \log \frac{p(Y,\mathcal{G}_p|\mathcal{G}_{sub})}{p(Y|\mathcal{G}_{sub})p(\mathcal{G}_p|\mathcal{G}_{sub})} \right].
\end{aligned}
\tag{20}
$$

By the definition of conditional probability, we have the following equation:

$$-I(Y;\mathcal{G}_{sub}) = -I(Y;\mathcal{G}_{sub},\mathcal{G}_p) + I(Y;\mathcal{G}_p|\mathcal{G}_{sub}). \tag{21}$$

Finally, we have the following equation by combining Eq. 18 and Eq. 21:

$$\min_{\mathcal{G}_{sub}} -I(Y;\mathcal{G}_{sub},\mathcal{G}_p) + I(Y;\mathcal{G}_p|\mathcal{G}_{sub}) + \beta I(\mathcal{G};\mathcal{G}_{sub}). \tag{22}$$

## A.2    Proof of Proposition 1

In this section, we provide a proof for Proposition 1 in the main paper.

For the term $I(Y;\mathcal{G}_p,\mathcal{G}_{sub})$,

$$
\begin{aligned}
I(Y;\mathcal{G}_p,\mathcal{G}_{sub}) &= \mathbb{E}_{Y,\mathcal{G}_{sub},\mathcal{G}_p} \left[ \log \frac{p\left(Y|\mathcal{G}_{sub},\mathcal{G}_p\right)}{p(Y)} \right] \\
&\geq \mathbb{E}_{Y,\mathcal{G}_{sub},\mathcal{G}_p} \left[ \log \frac{p\left(Y|\gamma(\mathcal{G}_{sub},\mathcal{G}_p)\right)}{p(Y)} \right].
\end{aligned}
\tag{23}
$$

We introduce a variational approximation $q_\theta\left(Y|\gamma\left(\mathcal{G}_{sub},\mathcal{G}_p\right)\right)$ of $p\left(Y|\gamma\left(\mathcal{G}_{sub},\mathcal{G}_p\right)\right)$.

$$
\begin{aligned}
I(Y;\mathcal{G}_p,\mathcal{G}_{sub}) &\geq \mathbb{E}_{Y,\mathcal{G}_{sub},\mathcal{G}_p} \left[ \log \frac{q_\theta\left(Y|\gamma(\mathcal{G}_{sub},\mathcal{G}_p)\right)}{p(Y)} \right] + \mathbb{E}_{Y,\mathcal{G}_{sub},\mathcal{G}_p} \left[ \log \frac{p\left(Y|\gamma(\mathcal{G}_{sub},\mathcal{G}_p)\right)}{q_\theta\left(Y|\gamma(\mathcal{G}_{sub},\mathcal{G}_p)\right)} \right] \\
&= \mathbb{E}_{Y,\mathcal{G}_{sub},\mathcal{G}_p} \left[ \log \frac{q_\theta\left(Y|\gamma(\mathcal{G}_{sub},\mathcal{G}_p)\right)}{p(Y)} \right] + \mathbb{E}_{\mathcal{G}_{sub},\mathcal{G}_p} \left[ KL\left[ p\left(Y|\gamma(\mathcal{G}_{sub},\mathcal{G}_p)\right) \| q_\theta\left(Y|\gamma(\mathcal{G}_{sub},\mathcal{G}_p)\right) \right] \right].
\end{aligned}
\tag{24}
$$

According to the non-negativity of KL divergence, we have:

$$I(Y; \mathcal{G}_p, \mathcal{G}_{sub}) \geq \mathbb{E}_{Y, \mathcal{G}_{sub}, \mathcal{G}_p} \left[ \log \frac{q_\theta \left( Y | \gamma(\mathcal{G}_{sub}, \mathcal{G}_p) \right)}{p(Y)} \right]$$
$$= \mathbb{E}_{Y, \mathcal{G}_{sub}, \mathcal{G}_p} \left[ \log q_\theta \left( Y | \gamma(\mathcal{G}_{sub}, \mathcal{G}_p) \right) \right] - \mathbb{E}_Y \left[ \log p(Y) \right] \quad (25)$$
$$= \mathbb{E}_{Y, \mathcal{G}_{sub}, \mathcal{G}_p} \left[ \log q_\theta \left( Y | \gamma(\mathcal{G}_{sub}, \mathcal{G}_p) \right) \right] + H(Y).$$

Finally, we have the following equation as:

$$I(Y; \mathcal{G}_p, \mathcal{G}_{sub}) \geq \mathbb{E}_{Y, \mathcal{G}_{sub}, \mathcal{G}_p} \left[ \log q_\theta \left( Y | \gamma(\mathcal{G}_{sub}, \mathcal{G}_p) \right) \right]. \quad (26)$$

### A.3 Decomposition of $I(Y; \mathcal{G}_p | \mathcal{G}_{sub})$ - Eq. 9

For the term $I(Y; \mathcal{G}_p | \mathcal{G}_{sub})$, by definition:

$$I(Y; \mathcal{G}_p | \mathcal{G}_{sub}) = \mathbb{E}_{Y, \mathcal{G}_{sub}, \mathcal{G}_p} \left[ \log \frac{p(Y, \mathcal{G}_p | \mathcal{G}_{sub})}{p(Y | \mathcal{G}_{sub}) p(\mathcal{G}_p | \mathcal{G}_{sub})} \right]. \quad (27)$$

By the definition of conditional probability, we have the following equation:

$$I(Y; \mathcal{G}_p | \mathcal{G}_{sub}) = \mathbb{E}_{Y, \mathcal{G}_{sub}, \mathcal{G}_p} \left[ \log \frac{p(Y, \mathcal{G}_{sub}, \mathcal{G}_p) p(\mathcal{G}_{sub})}{p(Y, \mathcal{G}_{sub}) p(\mathcal{G}_{sub}, \mathcal{G}_p)} \right]. \quad (28)$$

We allow the involvement of $\mathcal{G}_p$ in Eq. 28 as follows:

$$I(Y; \mathcal{G}_p | \mathcal{G}_{sub}) = \mathbb{E}_{Y, \mathcal{G}_{sub}, \mathcal{G}_p} \left[ \log \frac{p(Y, \mathcal{G}_{sub}, \mathcal{G}_p) p(\mathcal{G}_{sub}) p(\mathcal{G}_p)}{p(Y, \mathcal{G}_{sub}) p(\mathcal{G}_{sub}, \mathcal{G}_p) p(\mathcal{G}_p)} \right]$$
$$= \mathbb{E}_{Y, \mathcal{G}_{sub}, \mathcal{G}_p} \left[ \log \frac{p(Y, \mathcal{G}_{sub}, \mathcal{G}_p)}{p(Y, \mathcal{G}_{sub}) p(\mathcal{G}_p)} + \log \frac{p(\mathcal{G}_{sub}) p(\mathcal{G}_p)}{p(\mathcal{G}_{sub}, \mathcal{G}_p)} \right] \quad (29)$$
$$= \mathbb{E}_{Y, \mathcal{G}_{sub}, \mathcal{G}_p} \left[ \log \frac{p(Y, \mathcal{G}_{sub}, \mathcal{G}_p)}{p(Y, \mathcal{G}_{sub}) p(\mathcal{G}_p)} \right] - \mathbb{E}_{\mathcal{G}_{sub}, \mathcal{G}_p} \left[ \log \frac{p(\mathcal{G}_{sub}, \mathcal{G}_p)}{p(\mathcal{G}_{sub}) p(\mathcal{G}_p)} \right].$$

Finally, we have the following equation as:

$$I(Y; \mathcal{G}_p | \mathcal{G}_{sub}) = I(\mathcal{G}_p; Y, \mathcal{G}_{sub}) - I(\mathcal{G}_{sub}; \mathcal{G}_p). \quad (30)$$

### A.4 Additional Experiments

In this section, we present our additional experiments including ablation study (Section A.4.1), analysis of the different graph readout functions (Section A.4.2), reasoning process (Section A.4.3) and analysis of the hyperparameters $\alpha_1$, $\alpha_2$, $\alpha_3$ and $J$ (Section A.4.4). All of our experiments were performed with one NVIDIA GeForce A6000.

### A.4.1 Ablation Study

We perform ablation studies to examine the effectiveness of our model (i.e., PGIB and PGIB$_{\text{cont}}$). In Figure 7, the "*with all*" setting represents our final model that includes all the components. We conducted ablation studies on losses related to mutual information (i.e., $I(\mathcal{G}; \mathcal{G}_{sub})$ and $I(\mathcal{G}_{sub}; \mathcal{G}_p)$), merging prototypes, and the connectivity loss $\mathcal{L}_{\text{con}}$. We have the following observations: **1)** The performance of the models significantly deteriorates when the terms related to mutual information, $I(\mathcal{G}; \mathcal{G}_{sub})$ and $I(\mathcal{G}_{sub}; \mathcal{G}_p)$, are not considered, compared to our final model. Specifically, if we exclude the consideration of $\mathcal{G}_{sub}$ when constructing the prototypes $\mathcal{G}_p$ (i.e., without maximizing $I(\mathcal{G}_{sub}; \mathcal{G}_p)$), the representations of the prototypes that directly contribute to the final predictions cannot effectively obtain the informative information from the subgraph $\mathcal{G}_{sub}$, resulting in a deterioration of performance. Moreover, if we fail to incorporate the minimal sufficient information from the entire

graph $\mathcal{G}$ into $\mathcal{G}_{sub}$ (i.e., without minimizing $I(\mathcal{G}; \mathcal{G}_{sub})$), there is a higher likelihood of prototypes obtaining uninformative information, which can ultimately lead to a deterioration in performance. **2)** Merging prototypes improves both PGIB and PGIB$_{cont}$ by enhancing the distinguishability of the remaining prototypes. This process not only enhances the interpretability of the prototypes but also results in improved classification accuracy. By merging similar prototypes, important features are emphasized through the aggregation of weights from both prototypes. This results in a more precise and effective representation of the data, enhancing the model's interpretability and accuracy in classification performance (i.e., "*with all*" setting performs better than "*w/o merge*" setting.). **3)** The connectivity loss $\mathcal{L}_{con}$, which promotes the construction of more realistic subgraphs by inducing a compact topology, has a significant impact on the performance. This improvement can be attributed to the fact that subgraphs relevant to the target often form connected components in real-world datasets. Therefore, incorporating the connectivity loss leads to improved performance by ensuring that the subgraph maintains realistic connectivity patterns.

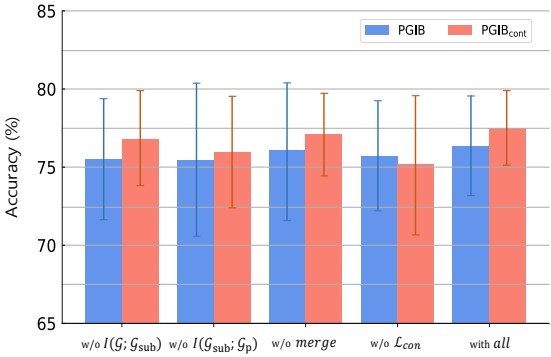

Figure 7: Ablation studies on PGIB.

### A.4.2 Analysis of the Different Graph Readout Functions

We conduct experiments on graph classification using different readout functions for PGIB. In Table 3, we show the classification performance based on three readout functions: max-pooling, mean-pooling, and sum-pooling, for both the PGIB and PGIB$_{cont}$. Table 3 demonstrates that max-pooling achieves the best performance in all datasets except for MUTAG dataset.

Table 3: Evaluation on the Different Graph Readout Functions (accuracy).

| Dataset | Methods | | | | | |
|---|---|---|---|---|---|---|
| | PGIB | | | PGIB$_{cont}$ | | |
| | MaxPool | MeanPool | SumPool | MaxPool | MeanPool | SumPool |
| MUTAG | 80.50 ± 7.07 | **86.50 ± 7.84** | 80.50 ± 10.39 | 85.50 ± 5.22 | **88.50 ± 6.34** | 86.50 ± 7.43 |
| PROTEINS | **77.14 ± 2.19** | 72.32 ± 5.17 | 60.89 ± 12.07 | **77.50 ± 2.42** | 68.39 ± 4.40 | 66.07 ± 4.79 |
| NCI1 | **77.65 ± 2.20** | 77.59 ± 7.41 | 63.96 ± 8.37 | **78.25 ± 2.13** | 77.52 ± 2.94 | 61.82 ± 3.96 |
| DD | **73.36 ± 1.80** | 67.56 ± 4.62 | 68.99 ± 4.56 | **73.70 ± 2.14** | 63.78 ± 5.40 | 64.12 ± 6.50 |

### A.4.3 Reasoning Process

We illustrate the reasoning process on two datasets, i.e., MUTAG and BA2Motif, in Figure 8. PGIB detects important subgraphs, and obtains similarity scores between subgraph $\mathcal{G}_{sub}$ and prototype $\mathcal{G}_p$. Then, PGIB computes the "points contributed" to predicting each class by multiplying the similarity score between $\mathcal{G}_{sub}$ and $\mathcal{G}_p$, with the weight assigned to each prototype in the prediction layer. Lastly, PGIB outputs the class with the highest total point among all the classes. We have the following observations in the reasoning process: **1)** PGIB identifies the specific substructures within $\mathcal{G}$ that contain label-relevant information by extracting $\mathcal{G}_{sub}$ from $\mathcal{G}$. **2)** PGIB identifies which training graphs play a crucial role in the predictions by conducting prototype projection to visualize the training graph that closely resembles the prototype. In other words, since each prototype is projected onto the nearest training graph, we can identify the training graph that had the most influence on predicting the target graph through the prototypes. **3)** PGIB identifies the influence of each "points contributed" on the final prediction by examining the total point to each class.

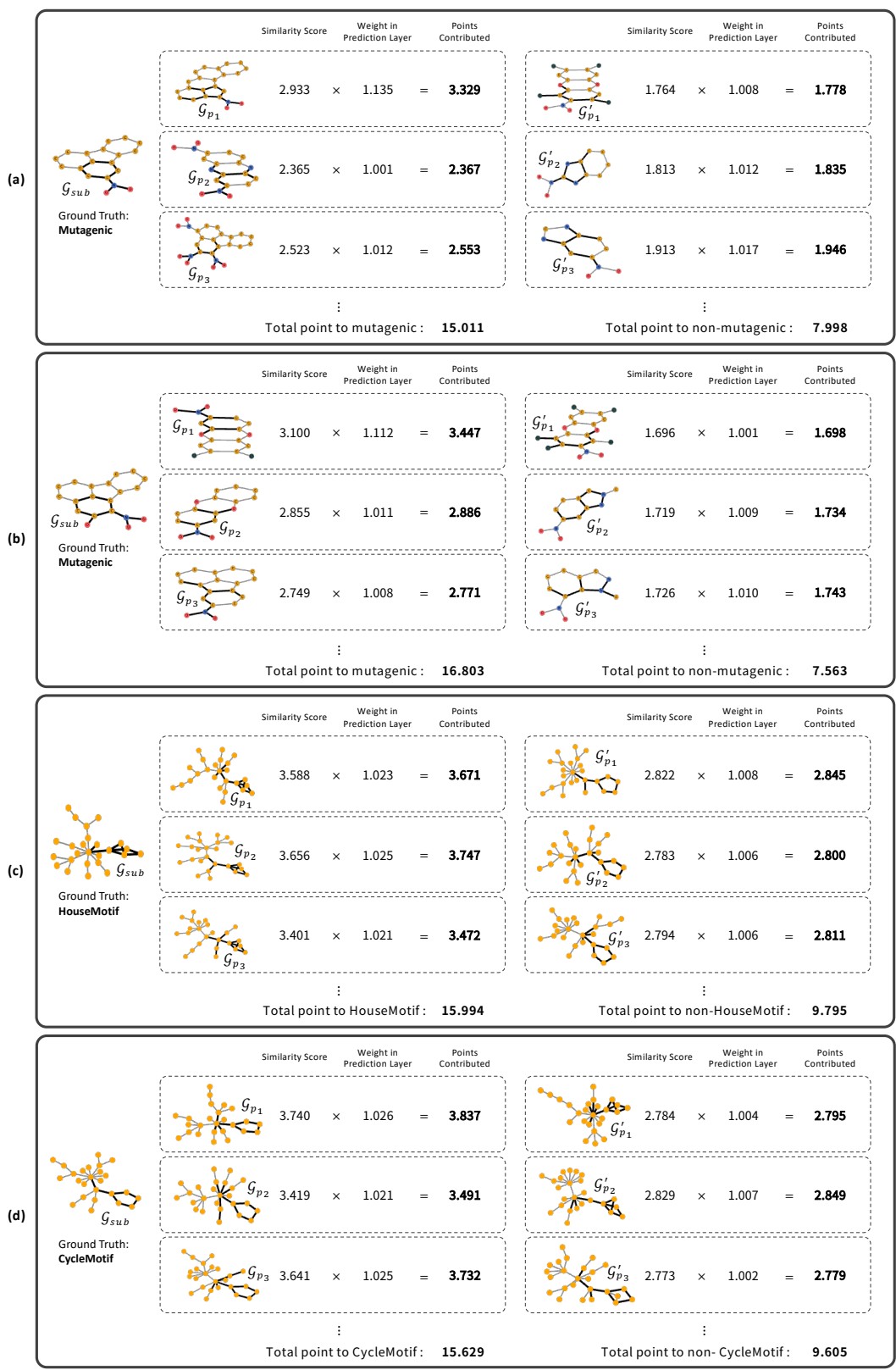

Figure 8: Reasoning process on MUTAG (a-b) and BA-2Motifs (c-d) datasets.

### A.4.4 Analysis of the Hyperparameters

We have conducted additional qualitative analysis. In this analysis, we compare the effects of different choices for $\alpha_1$, $\alpha_2$, $\alpha_3$ (i.e., loss weights) and $J$ (i.e., the number of prototypes for each class) at a more fine-grained level.

### A.4.4.1 Visualization of $\mathcal{G}_{sub}$ based on $\alpha_1$

Figure 6 shows a large value of $\alpha_1$ reduces the performance. For further analysis, we visualize the subgraph $\mathcal{G}_{sub}$ at different values of $\alpha_1$. The parameter $\alpha_1$ has an impact on the compression of the subgraph from the entire input graph. In Figure 9, when $\alpha_1$ is large, $\mathcal{G}_{\text{sub}}$ receives too much compressed information from $\mathcal{G}$, causing the loss of important data. It prevents $\mathcal{G}_p$ from containing label-relevant information, ultimately resulting in a negative impact on downstream performance.

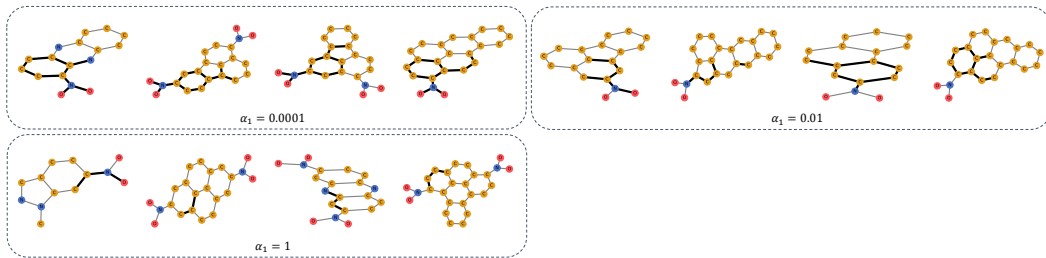

Figure 9: Visualization of $\mathcal{G}_{sub}$ based on $\alpha_1$ on MUTAG dataset

### A.4.4.2 Visualization of $\mathcal{G}_p$ based on $\alpha_2$

We have extended the scale of qualitative analysis on $\alpha_2$ in Figure 5 to provide a better understanding of its impact. It is crucial that the prototypes not only contain key structural information from the input graph but also ensure a certain level of diversity since each class is represented by multiple prototypes. In Figure 10, when we fix $\alpha_1$ at 0.1, the diversity of prototypes varies based on the degrees of $\alpha_2$. Specifically, when $\alpha_2$ becomes 1, the diversity of prototypes decreases, leading to a decline in the interpretability of the reasoning process and the overall model performance. This finding highlights the importance of selecting proper $\alpha_2$ to ensure both interpretability and performance are optimized.

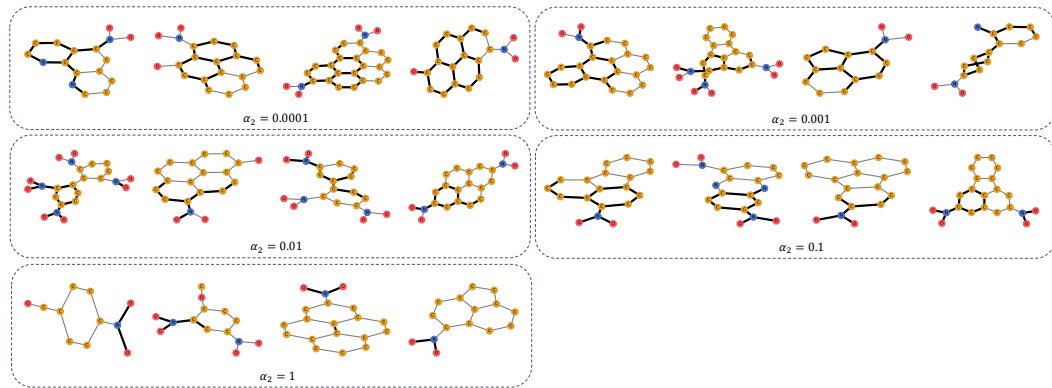

Figure 10: Visualization of $\mathcal{G}_p$ based on $\alpha_2$ on MUTAG dataset

### A.4.4.3 Visualization of $\mathcal{G}_{sub}$ based on $\alpha_3$

We mentioned that $\alpha_3$ is associated with the connectivity loss and plays a crucial role in influencing the interpretability of $\mathcal{G}_{sub}$ by promoting compact topology in Section 3.5. To verify this, we visualize the subgraph $\mathcal{G}_{sub}$ at different values of $\alpha_3$. In Figure 11, when we exclude the connectivity loss from the loss function (i.e., set $\alpha_3$ to 0), $\mathcal{G}_{sub}$ tends to consist of non-connected components. As a result, due to the wide and scattered range of detected subgraphs, the absence of connectivity loss results in the formation of unrealistic subgraphs.

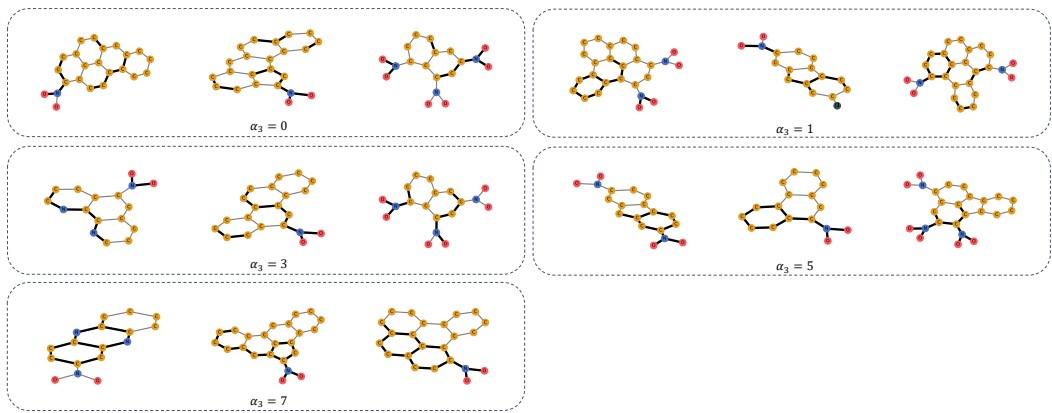

Figure 11: Visualization of $\mathcal{G}_{sub}$ based on $\alpha_3$ on MUTAG dataset

### A.4.4.4 Visualization of $\mathcal{G}_p$ based on $J$

We conducted interpretation visualizations of $\mathcal{G}_p$ based on the number of prototypes for each class in Figure 12. When the number of prototypes is small (as seen in the case with 3 prototypes), the prototypes do not contain diverse substructures. This limitation arises from making predictions using a restricted number of prototypes. On the other hand, if the number of prototypes is large (as shown in the case with 9 prototypes), a greater diversity of prototypes can be achieved because various and complex information can be obtained from $\mathcal{G}_{sub}$.

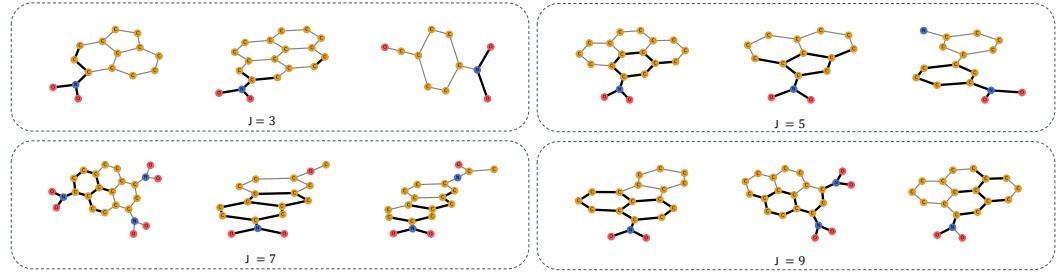

Figure 12: Visualization of $\mathcal{G}_p$ based on the number of prototypes ($J$) on MUTAG dataset

### A.5 Baselines

In this part, we provide details on the baselines used in our experiments.

- **ProtGNN** [35] utilizes prototypes to explain prediction results by potentially representing training graphs in graph neural networks. Specifically, ProtGNN measures the similarity between the embedding in the input graph and each prototype, and makes predictions based on the similarity. Additionally, ProtGNN can be extended to ProtGNN+ by incorporating a module that samples the subgraphs from the input graph to visualize the most similar subgraph to each prototype.

- **GIB** [31] utilizes the information bottleneck principle to detect important subgraphs in graph-structured data. Specifically, this method aims to extract subgraph embeddings by restricting the amount of information within the subgraph and retaining only the important information. During the training of graph neural networks, GIB encourages the recognition of important subgraphs within the graph data and performs graph classification tasks based on this recognition.

- **VGIB** [32] introduces noise injection into the graph information bottleneck. VGIB aims to enhance subgraph recognition by incorporating randomness into the graph data. This addition of randomness helps acquire diverse subgraph representations and captures the inherent uncertainty in the data, leading to enhanced performance in both classification tasks and subgraph recognition tasks.

- **GSAT** [13] aims to achieve interpretable and generalizable graph learning through a stochastic attention mechanism. It probabilistically estimates attention weights for the relationships between nodes and edges in a graph. The probabilistic attention mechanism enables the model to learn shared characteristics across domains and enhance its generalization performance.

- **GNNexplainer** [29] is a post-hoc interpretation model designed to interpret the prediction results of GNN models. Specifically, GNNExplainer optimizes a masking algorithm to maximize the mutual information with the existing label information. Its goal is to make the masked subgraph's prediction as close as possible to the original graph, which helps to detect substructures significant for predictions.

- **PGexplainer** [12] parameterizes the underlying structure as an edge distribution and generates the explanatory graph by sampling. PGExplainer collectively explains multiple instances by utilizing deep neural networks to parameterize the process of generating explanations. It enables the interpretability of the GNN model's behavior by adjusting the weight parameters of the GNN model, which allows it to be readily applied in an inductive setting.

Table 4: Source code links of the baseline methods

| Methods | Source code |
|---|---|
| ProtGNN | https://github.com/zaixizhang/ProtGNN |
| GIB | https://github.com/Samyu0304/graph-information-bottleneck-for-Subgraph-Recognition |
| VGIB | https://github.com/Samyu0304/Improving-Subgraph-Recognition-with-Variation-Graph-Information-Bottleneck-VGIB- |
| GSAT | https://github.com/Graph-COM/GSAT |
| GNNExplainer | https://github.com/RexYing/gnn-model-explainer |
| PGExplainer | https://github.com/flyingdoog/PGExplainer |

## A.6 Datasets

In this section, we provide details on the datasets used during training.

- **MUTAG** [16] consists of 188 molecular graphs, which are used to predict the properties of mutagenicity in chemical structures. The graph labels are determined by the mutagenicity of Salmonella typhimurium.

- **PROTEINS** [1] includes 1113 protein structures and is utilized for the classification of proteins into enzymes or non-enzyme. Each node represents an amino acid in the protein molecule, and edges connect nodes if the distance between the amino acids is less than 6 angstroms.

- **NCI1** [24] contains 4110 chemical compounds specifically designed for anticancer testing. Each chemical compound is labeled as either positive or negative based on its response to cell lung cancer.

- **DD** [5] consists of 1178 protein structures labeled as either an enzyme or a non-enzyme, similar to the previous dataset.

- **BA2Motifs** [12] is a synthetic dataset used for graph classification. Each graph is constructed based on a random graph generated using the Barabási–Albert (BA) model. It is then connected to one of two types of motifs: a house motif and a five-node cycle motif. The label of each graph is determined to belong to one of two classes based on the attached motif.

- **ZINC** [8] is a database of commercially accessible compounds used for virtual screening. It contains over 230 million purchasable compounds in a 3D format that can be docked readily.

## A.7 Limitations and Societal Impacts

PGIB does not incorporate domain knowledge, so domain-specific information cannot be conveyed to the extracted subgraph. For example, the extracted key subgraph may not necessarily correspond to a biologically or chemically existing functional group. It can cause unrealistic subgraphs to significantly affect the overall training of the model, including prototype training and final performance.

With the advancement and increasing sophistication of explainable artificial intelligence (XAI), these limitations may have a broader societal impact. There is a potential risk of excessive dependence

on XAI systems, leading to a decrease in human autonomy and decision-making. Blindly accepting the decisions of AI systems without critically evaluating XAI undermines human judgment and agency, which can potentially result in inappropriate or harmful behavior. For example, if a non-existent functional group is unquestioningly accepted from the model, it can lead to an erroneous understanding of the algorithm as a whole, with incorrect judgment about the functional group.

## A.8 Algorithm

---

**Algorithm 1:** Overview of PGIB Training

---

**Input:** Training dataset $\{(\mathcal{G}_i, y_i)\}_{i=1}^{n}$, prototype merge epoch $T_m$, prototype merge period $\tau$, the number of prototypes $M$, the number of classes $K$, hyper-parameters of the weights of the losses $\alpha_1, \alpha_2,$ and $\alpha_3$

1 **for** *training epochs* $t = 1, 2, \ldots, T$ **do**
2  $\quad \mathcal{G}_{sub} \leftarrow \arg\min_{\mathcal{G}_{sub}} I(\mathcal{G}; \mathcal{G}_{sub})$ by injecting noise into subgraph in Eq. 6  $\qquad$ // $\mathcal{L}_{\mathrm{MI}}^1$
3  $\quad$ Evaluate the loss $\mathcal{L}_{\mathrm{con}}$ in Eq. 15
4  $\quad r_m \leftarrow g(\mathbf{z}_{\mathcal{G}_{sub}}, \mathbf{z}_{\mathcal{G}_p}^m)$
5  $\quad$ Minimize $-I(\mathcal{G}_{sub}; \mathcal{G}_p)$ in Eq. 10 or 11  $\qquad$ // $\mathcal{L}_{\mathrm{MI}}^2$
6  $\quad$ Evaluate the loss $\mathcal{L}_{\mathrm{cls}} = -\frac{1}{N} \sum_{i=1}^{N} \sum_{c=1}^{K} \mathbb{I}(y_i = c) \log(\pi_c)$
7  $\quad$ **if** *Merge = True* **and** $t > T_m$ **and** $t\%\tau = 0$ **then**
8  $\quad\quad$ Calculate prototype similarity $h(\mathbf{z}_{\mathcal{G}_p^i}, \mathbf{z}_{\mathcal{G}_p^j}) = \left[ \sum_{\mathcal{G} \in \mathcal{X}} (g(\mathbf{z}_{\mathcal{G}_{sub}}, \mathbf{z}_{\mathcal{G}_p^i}) - g(\mathbf{z}_{\mathcal{G}_{sub}}, \mathbf{z}_{\mathcal{G}_p^j}))^2 \right]^{-1}$
9  $\quad\quad$ Perform prototype-merge
10 $\quad$ **end**
11 $\quad$ Total loss $\mathcal{L} = \mathcal{L}_{\mathrm{cls}} + \alpha_1 \mathcal{L}_{\mathrm{MI}}^1 + \alpha_2 \mathcal{L}_{\mathrm{MI}}^2 + \alpha_3 \mathcal{L}_{\mathrm{con}}$
12 $\quad$ Update model parameters by gradient descent
13 **end**

---