# OpenReview forum: "Interpretable Prototype-based Graph Information Bottleneck"
_NeurIPS.cc/2023/Conference — NeurIPS 2023 poster_

### Official Review · Reviewer_qf5C · 2023-06-14

**Soundness:** 3 good
**Presentation:** 3 good
**Contribution:** 2 fair
**Rating:** 5
**Confidence:** 4

**Summary:**

The authors propose a new GNN explanation framework, which combines prototype learning and the information bottleneck. They define the Prototype-based Graph Information Bottleneck framework in detail, and experiments have demonstrated the effectiveness of their proposed framework.

**Strengths:**

1.The authors introduce the Graph Information Bottleneck based on prototype learning, which makes a significant improvement over the baseline methods.

2.The experiments are comprehensive and there are many additional experiments in the appendix.

3.The paper is well written.




**Weaknesses:**

1.I argue that the novelty of the paper is limited by the fact that both the IB approach and the prototype approach have been well discussed before, and the authors seem to have simply combined the two methods together.

2.The number of prototypes per class is an important hyperparameter and the author should add a hyperparameter analysis of it.

3.I am not quite sure about the differences between PGIB and ProtGNN. If I use the GIB strategy in ProtGNN, what effect will it have?

4.In this paper, authors use GIN as a backbone. However, PGIB as a model-agnostic approach should be validated on more GNN structures, e.g. GCN, GraphSage.


Others:

1.Line 143 "PGIB s a novel " ->"PGIB is a novel "

2.Line 275 "This is because When" -> "This is because when"






**Questions:**

1.I wonder if the authors' proposed approach can alleviate the OOD problem which is widely discussed in the explanation GNNs[1,2].

2.I notice that the authors obtain two types of subgraphs, one in the Subgraph Extraction Layer and one in the Prototype Projection. In practice, which type of subgraph do we use as the final interpretation? What if the two types of subgraphs conflict?

3.If the interpretability of the prototypes is not considered,  can the Prototype Projection method  be removed?

References:

[1] Debiasing graph neural networks via learning disentangled causal substructure.

[2] Causal attention for interpretable and generalizable graph classiﬁcation.




**Limitations:**

See Weaknesses for details.

---

> ### Author Rebuttal · Authors · 2023-08-10
>
> **W1.**
> **# Method Selection Inspired by Motivation and Purpose**
> As highlighted in our introduction, ProtGNN utilizes prototypes to explain the model's training process (i.e., reasoning process (RP)). However, we found that ProtGNN overlooks the key substructures in input graphs, leading to less informative RP and decreased performance in downstream tasks (Fig. 1(a)). Drawing from this observation, we recognize the importance of including informative information from the input graphs, while excluding uninformative information in the RP.
> The key to addressing these challenges is through identifying the key substructure of an input graph. Among the methods available for extracting key substructures (e.g., GNNexplainer, PGexplainer, subgraphX, IB), we chose the IB-based approach because it is currently the most effective approach for identifying important substructures during the learning process. Unlike post-hoc methods, IB allows us to extract key substructures during the training process, aligning with our objectives. We would be delighted if the novelty of our work is recognized in the motivation behind our choices, rather than merely in the method itself.
> **# Theoretical Analysis for a Well-Designed Loss Function**
> Additionally, we want to clarify that our approach goes beyond a simple combination of existing methods. We designed and implemented our loss function based on a thorough theoretical analysis. Our work is the first paper to provide a theoretical framework for approaching IB from the perspective of prototypes. Specifically, we introduced a $L_{MI}^2$ loss, which not only improves the interpretability of the RP (See Fig 1(b)), but also the model performance on downstream tasks (See Fig 6(b)). The newly introduced loss, $L_{MI}^2$, plays a key role in enabling the mutual interaction between $G_{sub}$ obtained from the IB approach, and the prototype $G_p$.
> **# Interpretability Stabilization via Prototypes Merging**
> Lastly, it is important to note that, inspired by ProtoPShare, our work is the first work that introduces an effective method for merging prototypes in the graph domain, aimed at enhancing not only both the explanation of the reasoning process, but also the overall performance on downstream tasks.
> We hope that this clarification highlights the contributions of our research and provides a better understanding of our work's motivations and goals.
>
> **W2.**
> We have conducted additional experiments where we varied the number of prototypes per class. Table PDF-5 shows the results of the graph classification according to the number of prototypes.
> Additionally, we conducted interpretation visualizations of $G_p$ based on the number of prototypes in Figure PDF-4. When the number of prototypes is small, the prototypes do not contain diverse substructures. This limitation arises due to the necessity of making predictions using a restricted number of prototypes. On the other hand, if the number of prototypes is large, a greater diversity of prototypes can be achieved because various and complex information can be obtained from $G_{sub}$.
>
> **W3.**
> As mentioned in our response to W1, a naive combination of ProtGNN and GIB allows each to operate independently. As such, this simple combination does not consider the loss functions like $L_{MI}^2$ in PGIB, and thus, it leads to an uninformative reasoning process. This is because the prototypes may not adequately capture the key substructures that significantly influence the downstream label prediction. We want to emphasize that a naive combination of the prototype approach and the GIB approach is insufficient to enhance the interpretability of the reasoning process and the performance on downstream tasks; careful design of the losses is crucial.
>
> **W4.**
> We appreciate your valuable suggestions. In response, we conducted additional experiments using various GNN structures. The results are shown in the Table-P3. It demonstrates the strong performance of our model on different backbones.
>
> **Others**
> We appreciate your attention to the overlooked typos. We will make sure to fix them in the revised version.
>
> **Q1.**
> We sincerely appreciate your feedback, and we fully agree with your suggestion to include an OOD task. For these experiments, we utilized real-world molecule graphs, and followed existing studies [3,4] for OOD in which the data is split based on scaffold (i.e., scaffold split).
>
> [3] Li, Haoyang, et al. "Ood-gnn: Out-of-distribution generalized graph neural network." TKDE (2022).
> [4] Gui, Shurui, et al. "Good: A graph out-of-distribution benchmark." NIPS (2022).
>
> We conducted scaffold-based OOD experiments to evaluate the generalization performance of our model. We split our dataset based on the scaffold in a ratio of 8:1:1 and obtain a train and test dataset with totally different distribution by including the scaffold containing the most data in training set. Table-P4 is the result of the experiment, and it demonstrates the generalization performance of our model in an OOD setting.
>
> **Q2.**
> We would like to emphasize that explainability can be viewed from two perspectives: 1) interpretability of the model's predictions, and 2) interpretability of the model's reasoning process. As you mentioned, we obtain two types of subgraphs: a) from the Subgraph Extraction Layer, and b) from the Prototype Projection.
>
> - 'a)' provide interpretation for the model's predictions, giving insights into which subgraph is considered important when the model makes a final prediction.
> - 'b)' provide interpretation for the reasoning process. In other words, each class can be represented by the prototypes, and we can interpret how the model represents each class during the training process.
>
> We understand the reviewer’s confusion, and we will make sure to clearly explain them in the paper.
>
> **Q3.**
> Yes. If there is no need to interpret the reasoning process, it can be removed.

---

> > ### Comment · Reviewer_qf5C · 2023-08-11
> >
> > Thanks to the authors for the response. Although most of my questions have been addressed, I still have several questions. Specifically,
> >
> > 1.In Table PDF-3, authors compare PGB and PGB_cont with different network structures. However, I don't see the results for other baselines such as GSAT with GAT. I am not sure if PGB still maintains better performance with other network structures such as GAT.
> >
> > 2.I understand the authors' response to Q2. Can the author provide more specific examples to illustrate this process in detail? Authors can assume that PGB has been applied in practice, and what kind of explanation will PGB provide to enhance human's trust in PGB?

---

> > > ### Author Response · Authors · 2023-08-12
> > > **Response by authors**
> > >
> > > We appreciate your prompt response and valuable questions on our paper.
> > >
> > > 1.
> > > We have conducted additional experiments on other baselines with various backbones. The following table shows the results of classification performance
> > >
> > > |          |            |            |     GCN     |            |                |            |            |     GIN    |            |                |            |            |       GAT      |            |                |
> > > |:--------:|:----------:|:----------:|:-----------:|:----------:|:--------------:|:----------:|:----------:|:----------:|:----------:|:--------------:|:----------:|:----------:|:--------------:|:----------:|:--------------:|
> > > |          |   ProtGNN  |    GSAT    |     GIB     |    VGIB    |      PGIB      |   ProtGNN  |    GSAT    |     GIB    |    VGIB    |      PGIB      |   ProtGNN  |    GSAT    |       GIB      |    VGIB    |      PGIB      |
> > > |   MUTAG  | 85.00±4.47 | 79.00±9.17 | 75.50±10.36 | 70.50±7.89 | **86.00±3.74** | 80.50±9.07 | 80.00±8.94 | 79.00±6.24 | 81.00±6.63 | **82.50±3.54** | 86.00±3.74 | 80.00±7.75 |   81.50±9.23   | 67.00±8.12 | **86.50±3.91** |
> > > | PROTEINS | 71.96±1.34 | 70.45±3.10 |  70.80±6.93 | 75.54±5.04 | **77.86±1.48** | 73.83±4.22 | 69.64±4.71 | 75.25±5.92 | 73.66±3.32 | **77.50±2.42** | 72.14±1.54 | 68.21±3.37 |   75.45±3.42   | 70.09±6.47 | **78.21±1.04** |
> > > |    DD    | 65.29±2.38 | 71.34±3.95 |  72.35±7.15 | 65.46±6.27 | **72.69±1.14** | 69.15±4.33 | 71.93±2.74 | 72.61±8.26 | 68.32±6.20 | **73.70±2.14** | 65.21±1.60 | 71.76±3.28 | **73.53±7.49** | 66.89±3.13 |   71.34±1.95   |
> > >
> > > The reported results suggest that our method outperforms most other baselines in various backbones.
> > >
> > > ---
> > >
> > > 2.
> > > We will provide an example using a classification task on molecular graphs.
> > >
> > > Subgraph Extraction Layer presents to the user the important components (i.e., atoms) that had a significant impact on predicting the target molecule into the corresponding class. In other words, this explanation includes the subgraph structure of the atoms within the target molecule that play a significant role in the prediction.
> > >
> > > Prototype Projection provides the process through which the model predicts the target molecule into the corresponding class. The prototype shows the learned knowledge of the model (such as molecular structures learned during training time) that was utilized for predicting the target molecule. Specifically, since each prototype is projected onto the nearest training graph, we can identify the training graph that had the most influence on predicting the target molecule through the prototypes. For example, if the model's predicted probability for a specific molecule belonging to class c is high, it demonstrates to the user that the model considered a specific subgraph within a particular training molecule to be essential for class c, and that the prediction was made based on the presence of this subgraph within the target molecule. The purpose of such explanations is to present the model's reasoning process for molecular graphs in a way that is understandable to humans.
> > >
> > > As a result, PGIB aims to provide both interpretability from the perspective of input molecular graphs and the model's reasoning process, which can play a crucial role in enhancing the user's understanding and confidence in the model.
> > >
> > > We hope this explanation adequately addresses your question.
> > >
> > > We appreciate your quick response once again.

---

> > > > ### Comment · Reviewer_qf5C · 2023-08-12
> > > >
> > > > Thanks to the authors' reply, this Table is what I want to see. I will improve my score.

---

> > > > > ### Author Response · Authors · 2023-08-12
> > > > > **Author response**
> > > > >
> > > > > We thank the reviewer for acknowledging our effort, and for deciding to raise the score. We greatly appreciate it!

---

### Official Review · Reviewer_DMtB · 2023-06-24

**Soundness:** 2 fair
**Presentation:** 3 good
**Contribution:** 3 good
**Rating:** 6
**Confidence:** 4

**Summary:**

Interpretable graph learning can promote the use of graph-based scientific applications by providing model explanations. This paper focuses on extracting key subgraphs and employing a case-based reasoning process (also known as prototype learning) for model prediction. To make the extracted substructures more informative, the information bottleneck theory is incorporated into the subgraph extraction. Several strategies, such as prototype merging and connectivity constraints, are used to improve the model's interpretability and accuracy. Finally, this paper provides both qualitative and quantitative analyses of the model's interpretability and performance.

**Strengths:**

1.	The idea of using prototype learning to explain key substructures of the input graph is intuitive and reasonable.
2.	The extension of information bottleneck theory to prototype learning is inspiring and has practical utility in a wide range of application scenarios. In particular, it can help compress excessive information from the input graph and provide more fine-grained explanations, including key subgraphs and representative examples.
3.	The paper is well-organized and easy to follow.


**Weaknesses:**

1.	The related works are not properly cited. For example, the prototype merging method is quite similar to the one used in ProtoPShare [1], yet the author simply claims that "We propose a method to effectively merge the prototypes" (Line 224), without mentioning any existing works. This can confuse readers about the technical contribution.
[1] Rymarczyk, Dawid, et al. "Protopshare: Prototypical parts sharing for similarity discovery in interpretable image classification." Proceedings of the 27th ACM SIGKDD Conference on Knowledge Discovery & Data Mining. 2021.
2.	The paper's experiments are not sound enough. Firstly, the paper should include at least one black-box GNN model for performance prediction. Secondly, when evaluating the fidelity scores, ProtGNN should not be excluded (Section 4.3). Third, the paper is missing ablation studies for some proposed components, such as Connectivity loss.
3.	The design of dropping the first term in equation 9 (in Section 3.3.2) is not well-motivated. The goal is to minimize the whole $I(Y;\mathcal{G}p|\mathcal{G}{sub})$ instead of the single term $I(\mathcal{G}p ; \mathcal{G}{sub})$. The paper should provide essential theoretical analysis to better illustrate the effectiveness and rationality of this optimization goal.


**Questions:**

1.	What is the physical meaning of $p_i$ in Eq. 5? It seems to represent the extent of interference on the node representation. It is unclear how this practice can be considered as selecting a subgraph, making the physical meaning of connectivity loss also unclear.
2.	Why adding interference to the nodes according to the probability $p_i$ yields the representation of the subgraph? Is the subgraph representation obtained in this way equal to $f_g(G_{sub})$?
3.	In Eq. 8, how is the first inequality derived?


**Limitations:**

The authors have discussed the limitations and societal impacts.

---

> ### Author Rebuttal · Authors · 2023-08-10
>
> **W1** Thanks for raising this issue. In fact, we are aware of ProtoPShare, and thus we cited it in Sec 3.5 Line 223 reference “[15]”. While ProtoPShare is the first work that applies prototype merging in image classification, our work is the first work to demonstrate the effectiveness of prototype merging in graph-structured data. We fully agree that the sentence in Line 224 “We propose ... ” may have confused the reviewer, and we will make sure to revise along with an explicit citation: “Inspired by ProtoPShare “[15]”, we propose a prototype merging technique for graph-structured data..”
>
>  **W2**
> -**Black-Box GNN**
>
> Find results attached in pdf (Tab P1). We observe our model outperforms black-box GNNs.
>
> -**Fidelity Scores of ProtGNN**
>
> We would like to clarify that measuring the fidelity score of ProtGNN is not possible. We explain reasons:
> - First, recall that our model provides interpretability in two aspects: 1) model’s prediction based on the substructure of the input graph (i.e., $G_{sub}$) (Sec 4.3), and 2) model’s reasoning process based on the visualization of learned prototypes (i.e., $G_p$) (Apx 4.4).
> - Moreover, fidelity score quantifies the extent to which explanations accurately capture the important components that contribute to the model prediction (Line 307), meaning that fidelity score is measured based on $G_{sub}$ and not $G_p$.
> - However, ProtGNN is method for explaining the model’s reasoning process based on $G_p$, and it does not extract key subgraph $G_{sub}$. Hence, its fidelity score cannot be measured.
>
> We may still consider using $G_p$ to compute the fidelity score. However, as $G_p$ is obtained by projecting prototypes onto the training graphs, $G_p$‘s produced by different methods would be projected onto different training graphs, which means that different graphs are to be compared for same target. This variability makes it challenging to fairly compare fidelity scores across different graphs. Therefore, we only compare fidelity scores with methods that provide interpretability for $G_{sub}$ (GIB, VGIB, and PGIB).
>
> -**Ablation**
>
> Experiment on connectivity loss is shown in A.4.1. To provide further clarity in our analysis of PGIB, we conducted an ablation study including the connectivity loss at a more fine-grained level in Table R1, R2, and Table PDF-2. Additionally, interpretation visualization of $G_{sub}$ varying with Connectivity Loss is presented in Figure PDF-3.
>
> **W3**
> Note that the two terms in Eq 9 originate from $-I(Y;G_{sub})$, which is the 1st term in Eq 3. That is, minimizing $-I(Y;G_{sub})$ aims to increase the information regarding $Y$ in $G_{sub}$, and thus the two terms in Eq 9 both aim to increase the information regarding Y in $G_{sub}$.  Given this fact, it is intuitive that minimizing the 1st term in Eq 9, i.e., $I(G_p;Y,G_{sub})$, aims to reduce the information of Y from $G_p$, which eventually increases the information regarding Y in $G_{sub}$. In other words, $I(G_p;Y,G_{sub})$ includes removing the quantity of Y from $G_p$, because it solely aims to maximize $I(Y;G_{sub})$. However, our goal is to establish an interaction between $G_p$ and $G_{sub}$ to enhance both performance and interpretability of reasoning process. For this reason, we excluded the 1st term.
>
> **Q1**
>
> **-Eq.5**
> Eq. 5 controls the degree of interference in the node representation. $p_i$ is learned from node representation $h_i$ through MLP, and it attenuates the information of $G$ by injecting noise into node representation. ϵ is the noise sampled from parametric noise distribution. We assign each node a probability of being replaced by noise ϵ, and $p_i$ controls the information transmitted from $h_i$ and ϵ to $z_i$, e.g., when $p_i$ = 1, all information from $h_i$ is transmitted to $z_i$. Conversely, when $p_i = 0$, $z_i$ contains only noise ϵ and does not include any information from $h_i$. In short, we expect unimportant nodes to be replaced by noise ϵ, while remaining nodes constitute an important subgraph.
>
> **-Connectivity Loss**
> It aims to minimize the number of isolated nodes in $G_{sub}$, and promote connected subgraphs in order to enhance interpretability of $G_{sub}$. Specifically, $S_g$[j, 0] and $S_g$[j, 1] denote the probability of node $v_j$ ∈ $V_g$ belonging to $G_{sub}$ and $\bar{G}_{sub}$, respectively.
>
> $a_{11}=\sum_{i,j}A_{ij}p(V_i\in G_{sub}|V_i)p(V_j \in G_{sub}|V_j),$
> $a_{12}=\sum_{i,j}A_{ij}p(V_i\in G_{sub}|V_i)p(V_j \in \bar{G}_{sub}|V_j).$
>
> For example, we use $a_{11}$ and $a_{12}$ to denote the element (1,1) and the element (1,2) of $S^TAS$. Minimizing Lcon causes a11/(a11+a12) to approach 1, indicating that if Vi is in Gsub, its neighbors also have a high probability to be in $G_{sub}$. On the other hand, minimizing $L_{con}$ causes $\frac{a_{12}}{a_{11}+a_{12}}$ to approach 0, indicating that when $V_i$ is in $G_{sub}$, its neighbors have a low probability to be in $\bar{G}_{sub}$.
>
> In other words, $L_{con}$ aims to minimize the number of isolated nodes in a subgraph $G_{sub}$ by adjusting the node selection probabilities based on the connectivity information of $G$, which leads to stable connectivity.
>
> **Q2**
> Refer to our answer to Q1 for why adding interference yields the representation of the subgraph. Moreover, since we adopt the max pooling for $f_g$, the subgraph representation obtained in this way is equal to $f_g(G_{sub})$.
>
> **Q3**
> By definition we can derive,
>
> $I(Y;G_{sub},G_p) = E_ {Y,G_{sub},G_p}  [\log p(Y|G_{sub},G_p)-p(Y)]$,
>
> $I(Y;γ(G_{sub},G_p)) = E_{Y,G_{sub},G_p}  [\log p(Y| γ(G_{sub},G_p))-p(Y)]$.
>
> Additionally it is known that the following holds due to information loss in mutual information [1].
>
> $I(Y;G_{sub},G_p) ≥ I(Y; γ(G_{sub}, G_p))$
>
> Therefore, we can obtain following inequality:
>
> ${E}_ {Y,G_{sub},G_p}[\log p (Y |G_{sub}, G_p)] - E_Y [\log p(Y)] ≥ E_{Y,G_{sub},G_p} [\log p(Y |γ(G_{sub}, G_p))]−E_Y[\log p(Y )]$
>
> [1] Learning invariant graph representations for out-of-distribution generalization. NIPS22

---

> > ### Comment · Reviewer_DMtB · 2023-08-16
> >
> > Thank you for your detailed response. As my major concern is about missing experiments and the authors have added them, I'm happy to raise the score.

---

### Official Review · Reviewer_Q5w4 · 2023-07-06

**Soundness:** 2 fair
**Presentation:** 3 good
**Contribution:** 3 good
**Rating:** 6
**Confidence:** 3

**Summary:**

The paper introduces a new design of interpretable graph neural networks (GNNs) that combines ideas from information bottleneck approaches and prototype based approaches for by-design interpertation. The system titled PGIB extracts a subgraph from a given input graph and compares the embedding of the extracted subgraph in latent space to a set of learnt prototype subgraphs for final prediction. The authors evaluate their method on multiple datasets and baselines for classification performance and fidelity and demonstrate their efficacy for both.

**Strengths:**

1. The presented idea is novel. I have not seen any previous approach combining information bottleneck principles with prototype based classification.

2. The choice of baselines for comparison and overall experimental setup is solid. The main results in the paper are also convincing and positively reflect for the method.

3. The paper is generally well written with clear motivations.



**Weaknesses:**

1. The paper while reading gives sense of having too many moving parts.

2. Hyperparameter selection is understudied and feels arbitrary. The ablation studies should be strengthened.

3. Interpretation analysis --  Various choices in the system (loss weights or otherwise) are not well studied from the lens of interpretability. There are comments made around $\alpha_1, \alpha_2$ in Sec 4.4 but are not very well supported, with sparing analysis in appendix. At the very least for multiple choices you should include qualitative results specially for the ones for which you have some insights.

Overall it's a decent paper but various choices the authors make need to be better supported experimentally.

**Questions:**

1. **Hyperparameter selection:** -- I am a little puzzled by the hyperparameter choices selection. For eg. is the performance drop in Fig. 6(a) "drastic". It is just 2-3% drop in mean performance throughout from $\alpha_1=0$ to $1$. Am I understanding the graph incorrectly? The behaviour you highlight in A.4.3 is expected but I am rather thinking how is the performance still so high if the subgraphs are too compressed for $\alpha_1=1$? Similarly $\alpha_2$ choice also seems arbitrary based on the Fig. 6(b) with even less performance variation.

2. **Ablation** -- The number of prototypes and parameters for merge operation. How were they determined? How do they affect the interpretability and performance?

3. Typos in line 126 ("is maximizes"), line 312 ("Fidlity"), Fig 4 ('sparcity')

**Limitations:**

Authors discuss these in supplementary. I would prefer to have the limitations in the main paper.

---

> ### Author Rebuttal · Authors · 2023-08-09
>
> ---
>
> **Weaknesses**
>
> **W1.** Too many moving parts.
>
> **A1.** Thank you for your valuable feedback. We will make adjustments to the placement of the figures and tables to ensure they are easy to follow in our paper.
>
> ---
>
> **W2.** Hyperparameter selection / Ablation studies.
>
> **A2.**  We sincerely appreciate your feedback. To provide further clarity in our analysis of PGIB, we conducted an ablation study at a more fine-grained level than what was initially presented in the paper. We will include these additional results and analysis to provide a better understanding of our model. Here are the results and the corresponding analysis:
>
> **Table R1.**
> | **$\alpha_1$** |    **0**   | **0.0001** |  **0.001** |  **0.01**  |   **0.1**   |    **1**   |
> |:--------------:|:----------:|:----------:|:----------:|:----------:|:-----------:|:----------:|
> |    PROTEINS    |  76.8±3.16 | **77.50±2.42** | 75.72±2.32 | 75.54±3.04 |  73.75±5.37 | 73.48±4.25 |
> |      NCI1      | 75.21±1.41 | **78.25±2.13** | 73.75±3.04 | 74.09±3.15 |  72.55±2.55 | 69.09±2.18 |
>
> **Table R2.**
> | $\alpha_2$   | 0          | 0.0001         | 0.001      | 0.01             | 0.1        | 1          |
> |----------|------------|----------------|------------|------------------|------------|------------|
> | PROTEINS | 75.98±2.84 | 76.43±2.89 | 76.25±2.23 | 76.16±4.47       | **77.50±2.42** | 75.18±2.10 |
> | NCI1     | 74.79±1.59 | 76.16±2.92     | 75.84±2.91 | **78.25 ± 2.13** | 76.71±2.80 | 72.21±1.96 |
>
> Table R1 and R2 show that it is important to select appropriate parameters, as $\alpha_1$ has a significant influence on the degree of compression of the subgraph, and $\alpha_2$  plays an important role in learning the prototype, respectively.
>
> **Table R3.**
> | The number of prototypes | 3 → 1       | 4 → 2      | 5 → 3      | 6 → 4      | 7 → 5          | 8 → 6          | 9 → 7      |
> |--------------------------|-------------|------------|------------|------------|----------------|----------------|------------|
> | MUTAG                    | 81.5±3.91   | 82.0±6.4   | 84.5±4.7   | 85.0±5     | **85.5±5.22**  | 84.5±3.5       | 85.0±4.47  |
> | PROTEINS                 | 70.18.±2.19 | 71.7±3.02  | 74.4±2.11  | 75.26±2.82 | **77.5±2.42**  | 76.25±2.37     | 75.27±3.39 |
> | NCI1                     | 74.97±1.68  | 77.1±1.80  | 76.7±2.00  | 77.59±1.66 | 78.25±2.13     | **78.81±1.84** | 77.0±1.79  |
> | DD                       | 69.50±2.52  | 73.78±3.94 | 74.79±3.19 | 75.20±3.12 | **76.13±3.76** | 74.04±2.77     | 76.04±2.74 |
>
> Table R3 shows the importance of selecting a sufficient number of prototypes. For example, 4→2 indicates 4 prototypes per class, which are then merged to 2 prototypes per class. Since the number of prototypes determines the diversity of prototypes, a small number of prototypes hinders the formation of diverse prototypes.
>
> ---
>
> **W3.** Interpretation analysis
>
> **A3.**  Thank you for your valuable feedback. We have attached a pdf file containing interpretation analysis to the global response. We hope this response adequately addresses your concern.
>
> ---
>
> **Questions**
>
> **Q1.** Hyperparameter selection
>
> **A.** Thank you for sharing your insights on our experiments. To further clarify our analysis of Figure 6 and A.4.3, we have extended the analysis for $\alpha_1$ and $\alpha_2$ and also included performance evaluation using additional dataset.
>
> - Regarding the term “drastic” in Figure 6(a)
>   -  We understand that it might be subjective. We want to emphasize that similarly to other GIB methods [1,2], the overall performance of our method is primarily influenced by the classification loss. This explains why the performance remains relatively high even when the value of $\alpha_1$ is high. In the context of $\alpha_1$, the term "drastic" refers to the relative impact on the performance within a specific range of $\alpha_1$ values. We have observed a similar tendency in the results of additional dataset as well in Table R1.
>
> - Regrading the Less Variation in Performance in Figure 6(b)
>   -  As we mentioned before, the overall performance of our method is mainly affected by the classification loss. To conduct a more comprehensive analysis of the performance variation influenced by $\alpha_2$, we conducted experiments on an additional dataset in Table R2. These results demonstrate that although it exhibits less variation in performance compared to the effect of $\alpha_1$, it still clearly illustrates the performance gap between the model with an appropriate $\alpha_2$ and the model without it.
>   -  Furthermore, we want to emphasize that $\alpha_2$ significantly impacts the interpretability of the reasoning process in the Figure PDF-1. As we addressed in the global response, $\alpha_2$ plays a crucial role in ensuring the interpretability of prototypes, thereby enhancing our model's overall explainability.
>
> **Q2.**  Ablation
>
> **A.**  Thank you for raising this important point and for your valuable suggestion. As mentioned earlier, in Table R3, we have conducted additional experiments where we varied the number of prototypes per class and the final number of prototypes after merging them.
> Additionally, we conducted interpretation visualizations of $\mathcal{G}_ {p}$ based on the number of prototypes in Figure PDF-4. When the number of prototypes is small, the prototypes do not contain diverse substructures. This limitation arises due to the necessity of making predictions using a restricted number of prototypes. On the other hand, if the number of prototypes is large, a greater diversity of prototypes can be achieved because various and complex information can be obtained from $\mathcal{G}_ {sub}$.
>
> [1] Yu, Junchi, et al. "Graph information bottleneck for subgraph recognition." arXiv 2020
>
> [2] Yu, Junchi, et al. "Improving subgraph recognition with variational graph information bottleneck." CVPR 2022
>
> **Q3.** Typos
>
> We appreciate your attention to the overlooked typos. We will make sure to fix them for the revised version.

---

> > ### Comment · Reviewer_Q5w4 · 2023-08-13
> > **Rebuttal acknowledgement**
> >
> > Thank you for the rebuttal!
> >
> > I also want to request the authors if the can add curves for training loss or validation/test performance in the appendix for different hyperparameter settings. I am interested to see how the loss functions optimize and if some balance between $\alpha_1, \alpha_2, \alpha_3$ can also be achieved besides tracking the performance.
> >
> > In light of the new experiments, and analysis, I'd like to raise my score from 5 to 6.

---

> > > ### Author Response · Authors · 2023-08-16
> > > **Response by authors**
> > >
> > > We thank the reviewer for acknowledging our effort, and for deciding to raise the score.
> > >
> > > We provide our test performances according to $\alpha_1$, $\alpha_2$, and $\alpha_3$ in our model. We present the performance based on a few epochs in a simple table format, since we cannot provide you with figures in the discussion phase.
> > >
> > > |           |        |            | **$\alpha_1$** |          |         |        |           |        |            | **$\alpha_2$** |          |         |        |           |        |        | **$\alpha_3$** |        |        |
> > > |:---------:|:------:|:----------:|:--------------:|:--------:|:-------:|:------:|:---------:|:------:|:----------:|:--------------:|:--------:|:-------:|:------:|:---------:|:------:|:------:|:--------------:|:------:|:------:|
> > > | Epoch |  **0** | **0.0001** |    **0.001**   | **0.01** | **0.1** |  **1** | Epoch |  **0** | **0.0001** |    **0.001**   | **0.01** | **0.1** |  **1** | Epoch |  **0** |  **1** |      **3**     |  **5** |  **7** |
> > > |     1     | 0.4414 |   0.4414   |     0.4685     |  0.4685  |  0.4865 | 0.4775 |     1     | 0.4505 |   0.4414   |     0.4505     |  0.4414  |  0.4414 | 0.4595 |     1     | 0.3604 | 0.3694 |     0.3604     | 0.3694 | 0.3604 |
> > > |     10    | 0.6036 |   0.6036   |     0.5856     |  0.4775  |  0.5315 | 0.4234 |     5     | 0.4595 |   0.4234   |     0.4685     |  0.4324  |  0.4955 | 0.4505 |     5     | 0.4865 | 0.4955 |     0.4775     | 0.4685 | 0.4775 |
> > > |     20    | 0.6577 |   0.6667   |     0.6667     |  0.5946  |  0.6036 | 0.5676 |     10    | 0.5676 |   0.5766   |     0.6036     |  0.5856  |  0.5946 | 0.5856 |     10    | 0.5676 | 0.5856 |     0.5856     | 0.5946 | 0.5495 |
> > > |     25    | 0.7117 |   0.7207   |     0.7207     |  0.6667  |  0.6396 | 0.5676 |     15    | 0.5766 |   0.6216   |     0.6036     |  0.6216  |  0.6216 | 0.6036 |     15    | 0.5856 | 0.6126 |     0.5766     | 0.6036 | 0.5856 |
> > > |     35    | 0.7477 |   0.7477   |     0.7207     |  0.7117  |  0.6577 | 0.5135 |     25    | 0.6757 |   0.6757   |     0.6757     |  0.6667  |  0.7027 | 0.6667 |     20    | 0.6847 | 0.6577 |     0.6937     | 0.7207 | 0.6847 |
> > > |     45    | 0.7477 |   0.7658   |     0.7207     |  0.7387  |  0.6126 | 0.5495 |     30    | 0.6937 |   0.6937   |     0.7297     |  0.6847  |  0.7387 | 0.7117 |     35    | 0.6757 | 0.6937 |     0.6847     | 0.7387 | 0.7207 |
> > > |     50    | 0.7568 |   0.7568   |     0.7207     |  0.6757  |  0.6577 | 0.5405 |     35    | 0.6937 |   0.7027   |     0.7027     |  0.7027  |  0.7477 | 0.7207 |     40    | 0.7117 | 0.7477 |     0.7387     | 0.7477 | 0.7207 |
> > > |     55    | 0.7387 |   0.7928   |     0.7568     |  0.6667  |  0.6306 | 0.5315 |     40    | 0.7387 |   0.7658   |     0.7477     |  0.7297  |  0.7748 | 0.7568 |     50    | 0.7207 | 0.7387 |     0.7568     | 0.7658 | 0.7477 |
> > >
> > > We will consider your suggestion to include more detailed curves for various hyperparameters in the revised version of the appendix. We have discovered the optimal values of $\alpha_1$, $\alpha_2$, and $\alpha_3$ through various hyperparameter searches to achieve balance.
> > >
> > > We appreciate your valuable suggestions on our paper.

---

### Official Review · Reviewer_fJjc · 2023-07-10

**Soundness:** 3 good
**Presentation:** 4 excellent
**Contribution:** 3 good
**Rating:** 6
**Confidence:** 3

**Summary:**

This paper investigates the usage of prototype learning for GNN explainability, focusing in particular on identifying key subgraphs through the graph information bottleneck principle.

Extensive experiments are conducted, which consider several baselines and different molecular datasets.

**Strengths:**

- The scope of the paper is well-defined, and the goal of the novel method is relevant.
- The method is well described and motivated, and the code has been made available to improve reproducibility.
- The baselines include diverse, recent SOTA methods. Different real-world datasets have been used. Overall, the experiments appear robust.

**Weaknesses:**

- The main weakness in my opinion is that it is hard to conclude that the proposed method consistently outperforms all the baselines across all the metrics, given the reported results. For example, are the results in Table 1 statistically significant, given how large the confidence interval is? Similarly, in Table 2 for F+, VGIB reports the same results (why they are not bold?).
I think that the proposed method clearly has an advantage for some properties/metrics, but the complexity of the results should be better discussed n the text. I suggest (1) extending the discussion about the limitations of the model and (2) extending the analysis to investigate in what contexts (and why) the model significantly outperforms the baseline.

**Questions:**

See previous point.

**Limitations:**

See previous point on better discussing the limitations of the results.

---

> ### Author Rebuttal · Authors · 2023-08-09
>
> ---
> **Weaknesses**
>
> **Q1.** The main weakness in my opinion is that it is hard to conclude that the proposed method consistently outperforms all the baselines across all the metrics, given the reported results. For example, are the results in Table 1 statistically significant, given how large the confidence interval is? Similarly, in Table 2 for F+, VGIB reports the same results (why they are not bold?).
>
> **A.** Thank you for your valuable feedback. To address your concern, we have conducted a statistical analysis, specifically a paired t-test (n=10), providing the p-value and the confidence interval. We perform a t-test between our methods and the runner-up baseline. Here are the results of the experiments.
>
> ---
> | Table 1 (Paired t-test) |          |         |             |
> |:-----------------------:|:--------:|:-------:|:-----------:|
> |         Dataset         | baseline | p-value |      CI     |
> |          MUTAG          |   VGIB   |  0.0606 |  (0.0, inf) |
> |         PROTEINS        |    GIB   |  0.0080 | (0.01, inf) |
> |           NCI1          |  ProtGNN |  0.0006 | (0.02, inf) |
> |            DD           |   VGIB   |  0.0962 |   0.0962   |
>
> We established a confidence level of 95% and an alternative hypothesis indicating that the mean differences between our method (PGIBcont) and the baseline are greater than 0. With the exception of MUTAG and DD datasets, the p-values are sufficiently small to reject the null hypothesis, while the two datasets (MUTAG and DD) still exhibit values that are relatively close to the significance level. For the MUTAG dataset, which inherently contains a small number of data points (188 instances), the outcomes tend to exhibit a larger standard deviation (as indicated by [1], [2] ). In DD dataset, IB based methods tend to have a large standard deviation. However, we would like to highlight that our method does not exhibit such tendencies even if it is based on IB. We attribute this to the consideration of various prototypes beyond merely the substructures of the input graphs.
> Regarding Table 2 for F+, we apologize for any confusion that we may have caused. Thank you for bringing this to our attention. We will make sure to fix them for the revised version.
>
> [1] Sui, Y., Wang, X., Wu, J., Lin, M., He, X., & Chua, T. S. Causal attention for interpretable and generalizable graph classification. KDD 2022
>
> [2] Yu, J., Xu, T., Rong, Y., Bian, Y., Huang, J., & He, R. (2020). Graph information bottleneck for subgraph recognition. arXiv preprint arXiv:2010.05563.
>
> ---
>
> **Q2.** I think that the proposed method clearly has an advantage for some properties/metrics, but the complexity of the results should be better discussed n the text. I suggest (1) extending the discussion about the limitations of the model and (2) extending the analysis to investigate in what contexts (and why) the model significantly outperforms the baseline.
>
> **A.**  Thank you for your valuable suggestion regarding further discussions on advantages and limitations of our proposed method.
> - **A further limitation of our model**: Beyond the limitation we mentioned in A.7, the performance of our proposed method can be compromised when graphs in the dataset inherently lack task-relevant substructures (e.g., social network data, traffic network data, etc). This is mainly because the key underlying assumption of our model is the existence of task-relevant important substructures.
> - **When and why our method performs well**: Conversely, our proposed method demonstrates strong performance when the dataset comprises explicit label-representing substructures, as seen in molecular graphs, due to its capacity to capture subgraphs and conduct predictions. Additionally, the complex correlation between the substructure and various prototypes can be actively used for label prediction, which can improve both label prediction performance and interpretability.
>
> We thank the reviewer for the valuable suggestion, and we will make sure to include the above discussions in the revised paper.

---

> > ### Comment · Reviewer_fJjc · 2023-08-20
> > **Response to rebuttal**
> >
> > Thanks for the response, which helps clarify my questions.
> > Overall, I confirm my acceptance rating.

---

### Author Rebuttal · Authors · 2023-08-09

We appreciate the reviewers for their valuable comments on our paper. We have conducted additional qualitative analysis, which is provided in the attached PDF. In this analysis, we compare the effects of different choices for $\alpha_1$, $\alpha_2$, $\alpha_3$ (i.e., loss weights) and the number of prototypes at a more fine-grained level.
- Impact of $\alpha_1$

As discussed in Section 4.4, the hyperparameter $\alpha_1$ has an impact on the compression of the subgraph from the entire input graph. In Figure PDF-2, when $\alpha_1$ is less than 0.01, certain parts of the key substructure, specifically NO2 in this case, are excluded, resulting in a decrease in both interpretability and performance.
- Impact of $\alpha_2$

We have extended the scale of qualitative analysis on $\alpha_2$ shown in Figure 5 to provide a better understanding of its impact. It is crucial that the subgraphs of prototypes not only contain key structural information from the subgraph found by IB but also ensure a certain level of diversity, since each class is represented by multiple prototypes to enhance the model's capacity.
In Figure PDF-1, when we fix $\alpha_1$ to 0.1 the diversity of prototypes varies based on the degrees of $\alpha_2$. Specifically, when $\alpha_2$ becomes 1, the diversity of prototypes decreases, leading to a decline in the interpretability of the reasoning process and the overall model performance. This finding highlights the importance of selecting proper $\alpha_2$  to ensure both interpretability and performance are optimized.
- Impact of $\alpha_3$

The hyperparameter $\alpha_3$ is associated with the connectivity loss, which plays a crucial role in the interpretability of $G_{sub}$ by promoting a compact topology. In real-world datasets, the key substructure often tends to form non-connected components without $\alpha_3$. In Figure PDF-3, when we exclude the connectivity loss from the final loss function (i.e., set $\alpha_3$ to 0), $G_{sub}$ tends to consist of multiple connected components. As a result, due to the wide and scattered range of detected subgraphs, the absence of connectivity loss results in the formation of unrealistic subgraphs.

We hope that this material highlights the contributions of our research and provides a better understanding of our work.

---

### Decision · Program_Chairs · 2023-09-21

**Decision:**

Accept (poster)

**Comment:**

Reviewers found this paper exciting and unanimously recommended accept.